# Prediction-Powered Risk Monitoring of Deployed Models for Detecting Harmful Distribution Shifts

**Guangyi Zhang** [1]  **Yunlong Cai** [1]  **Guanding Yu** [1]  **Osvaldo Simeone** [2]

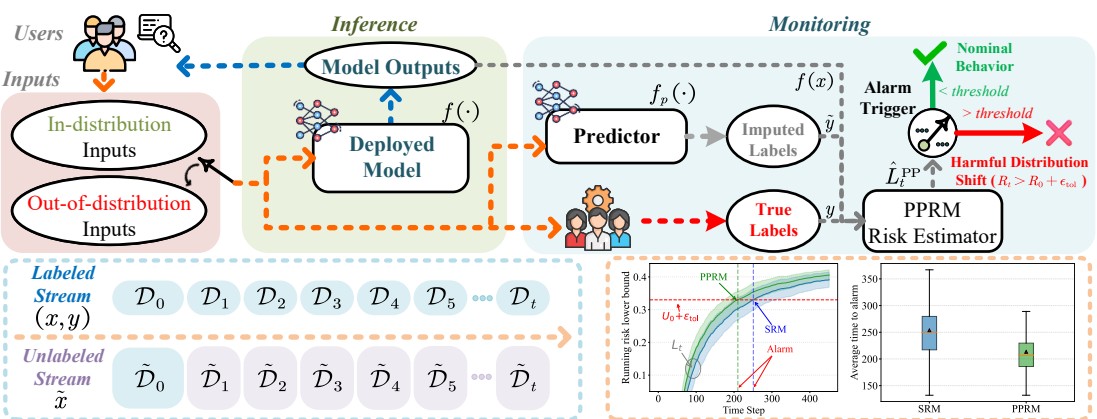

*Figure 1.* Over a discrete-time index, $t = 1, 2, ...$, a deployed system is monitored to detect harmful data distribution shifts that cause the running risk $\bar{R}_t$ to exceed the nominal risk $R_0$ by more than a maximum tolerated value $\epsilon_{\text{tol}}$. Supervised risk monitoring (SRM) assumes access to a labeled dataset $\mathcal{D}_t$ for $t = 0, 1, ...$ (with $t = 0$ corresponding to nominal behavior) of the form $(x, y)$, where $x$ is the input and $y$ is the true label. In contrast, the proposed prediction-powered risk monitoring (PPRM) leverages both labeled data and unlabeled data $\tilde{\mathcal{D}}_1, \tilde{\mathcal{D}}_2, ...$, including inputs $\tilde{x}$ only, by integrating an auxiliary pre-designed function $f_{\text{p}}(\cdot)$, yielding synthetic labels $\tilde{y}$.

## Abstract

We study the problem of monitoring model performance in dynamic environments where labeled data are limited. To this end, we propose prediction-powered risk monitoring (PPRM), a semi-supervised risk-monitoring approach based on prediction-powered inference (PPI). PPRM constructs anytime-valid lower bounds on the running risk by combining synthetic labels with a small set of true labels. Harmful shifts are detected via a threshold-based comparison with an upper bound on the nominal risk, satisfying assumption-free finite-sample guarantees on the type-I error. We demonstrate the effectiveness of PPRM through extensive experiments on image classification, large language model (LLM), and

telecommunications monitoring tasks.

## 1. Introduction

Modern machine learning models have been widely deployed in dynamic real-world environments where the underlying data distribution may shift over time (Zawalski et al., 2025). In such scenarios, systems can experience performance degradation due to factors such as changed environmental conditions (Lipton et al., 2018), as well as evolving user preferences (Guan et al., 2025). This raises concerns about reliability and safety, particularly in critical applications such as autonomous driving (Sun et al., 2025), medical diagnosis (Zeb et al., 2024), and engineering (Simeone et al., 2026). Consequently, there is a pressing need to continuously monitor the performance of deployed models.

Once a monitoring system raises an alarm, the deployed machine learning model may be deemed to require retraining or substitution. Given the cost associated with such extreme measures, recent works have suggested designing sequential testing frameworks that offer formal type-I error guarantees (Podkopaev & Ramdas, 2022; Amoukou et al., 2024; Schirmer et al., 2025). As shown in Figure 1, these methods

[1]College of Information Science and Electronic Engineering, Zhejiang University, Hangzhou 310027, China [2]Institute for Intelligent Networked Systems (INSI), Northeastern University London, One Portsoken Street, London E1 8PH, United Kingdom. Correspondence to: Yunlong Cai <ylcai@zju.edu.cn>.

*Proceedings of the 43$^{rd}$ International Conference on Machine Learning*, Seoul, South Korea. PMLR 306, 2026. Copyright 2026 by the author(s).

aim to raise an alarm when the time-averaged accumulated expected risk exceeds a predefined threshold. The initial work (Podkopaev & Ramdas, 2022) introduced *supervised risk monitoring* (SRM), which is based on time-uniform concentration bounds obtained from labeled calibration data. Thanks to the anytime-valid properties of such bounds, SRM guarantees assumption-free type-I error requirements.

However, labeled data from deployed models are often scarce and costly to obtain, making conventional fully supervised monitoring challenging. Based on this observation, Amoukou et al. introduced purely unsupervised monitoring methodologies, which do not require labels for monitoring. However, in order to provide type-I error guarantees, purely unsupervised methods require assumptions about the calibration of the deployed model, which is used to impute missing labels. These assumptions are difficult to validate. Furthermore, unsupervised approaches typically require a large number of samples to achieve reliable detection, which may not be practical in many real-world settings.

To address these challenges, we propose *prediction-powered risk monitoring* (PPRM), a framework that generalizes SRM to the semi-supervised setting based on *prediction-powered inference* (PPI) (Angelopoulos et al., 2023a). As illustrated in Figure 1, PPRM assumes the availability of both labeled data and unlabeled data at each monitoring round. Following PPI, PPRM utilizes predictions on unlabeled data as synthetic labels, while correcting the resulting bias using a small set of labeled samples. In this way, PPRM can provide risk estimates with lower variance, and thus produces tighter confidence bounds than SRM. Therefore, as shown at the bottom of Figure 1, the lower bound can exceed the threshold earlier, while also providing a similar type-I error guarantee.

We illustrate our procedure in three tasks, including image classification, large language model (LLM) monitoring via an LLM-as-a-judge (Zheng et al., 2023), and a channel equalization task in a telecommunication system (Zecchin et al., 2023).

Our main contributions are as follows:

- We introduce PPRM, a semi-supervised extension of SRM that combines synthetic labels from unlabeled data with a small number of true labels through PPI.

- We prove that PPRM provides rigorous assumption-free statistical guarantees on the type-I error.

- We further develop an online adaptive procedure to adjust the reliance on unlabeled data, improving efficiency while preserving statistical validity.

- We validate our methods through multiple case studies, including LLM monitoring via an LLM-as-a-judge,

showing that PPRM achieves more accurate and timely risk monitoring compared to existing supervised and unsupervised methods.

## 2. Problem Formulation

In this section, we formalize the problem of monitoring model performance over time with the aim of identifying harmful distribution shifts that cause an excessive degradation in performance. We also review the benchmark approach introduced in (Podkopaev & Ramdas, 2022).

### 2.1. Setting and Problem Formulation

As illustrated in Figure 1, we focus on a standard multiclass classification framework, where the input domain is represented by a set $\mathcal{X} \subseteq \mathbb{R}^D$ and the output label space is $\mathcal{Y} = \{1, \ldots, C\}$, with $C$ denoting the total number of classes. We study the problem of monitoring the performance of a deployed model $f(x) \in \mathcal{O}$, while processing test data sequentially. The output space $\mathcal{O}$ refers to the label space $\mathcal{Y}$ or to the space of probability distributions over $\mathcal{Y}$, depending on whether the prediction is deterministic or probabilistic. We fix a loss function $\ell : \mathcal{O} \times \mathcal{Y} \to \mathbb{R}$, which is assumed to be bounded in the interval $[0, 1]$.

Each data point $(x, y)$ corresponds to an independent realization from an unknown joint probability distribution over the set $\mathcal{X} \times \mathcal{Y}$. *Prior* to deployment, we assume access to a *calibration dataset*, whose data points are assumed to be sampled independently and identically from a fixed *source distribution* $P_0$. This distribution represents nominal conditions under which the deployed model $f(x)$ is known to perform satisfactorily. For instance, for a robotics application, calibration data may be collected in environments for which the robot has been extensively trained and validated.

At *deployment time*, test data are received *sequentially* over a discrete time index $t = 1, 2, \ldots$, and are assumed to be sampled from a potentially non-stationary *test* distribution, denoted by $P_t$ for $t = 1, 2, \ldots$ No specific assumptions are imposed on the distributional shifts encoded by the sequence $\{P_t\}_{t \geq 1}$ over time. The goal is to quickly detect if the distributions $P_t$ are sufficiently distinct from the nominal distribution $P_0$ to cause a harmful shift, translating into an excessive degradation in model performance. Throughout this paper, the notation $(x_t, y_t) \sim P_t$ represents a generic input-output pair drawn from the distribution $P_t$, whereas the pair $(x_{t,i}, y_{t,i})$ corresponds to the $i$-th observed realization in the dataset $\mathcal{D}_t$.

In more detail, the calibration dataset $\mathcal{D}_0 = \{(x_{0,i}, y_{0,i})\}_{i=1}^{n_0}$ is assumed to be drawn i.i.d. from the source distribution $P_0$. Furthermore, at each discrete test time $t$, a dataset $\mathcal{D}_t = \{(x_{t,i}, y_{t,i})\}_{i=1}^{n_t}$ is obtained whose samples $(x_{t,i}, y_{t,i})$ are drawn i.i.d. from the test

distribution $P_t$. The expected loss of the model $f(\cdot)$ at time $t$ is given by

$$R_t = \mathbb{E}\left[\ell\big(f(x_t), y_t\big)\right], \tag{1}$$

with $(x_t, y_t) \sim P_t$, for $t = 0, 1, ...$

The quantity $R_0$ corresponds to the *source risk*, i.e., to the expected loss when inputs are drawn from the source distribution $P_0$. As mentioned, this serves as a benchmark nominal risk, and excessive positive deviations from it signal a harmful distribution shift.

The test performance is evaluated by the *running test risk* (Podkopaev & Ramdas, 2022)

$$\bar{R}_t = \frac{1}{t} \sum_{t'=1}^{t} R_{t'}, \tag{2}$$

which measures the average test risk of the model across the first $t$ test time steps. If, at some time point $t^*$, the running risk $\bar{R}_{t^*}$ exceeds the baseline source risk $R_0$ by a pre-defined tolerance parameter $\epsilon_{\text{tol}} > 0$, i.e., if

$$\bar{R}_{t^*} > R_0 + \epsilon_{\text{tol}}, \tag{3}$$

the risk monitor should ideally raise an alarm, indicating that deployment may be stopped and retraining or fine-tuning may be initiated.

Accordingly, *risk monitoring* addresses the binary hypothesis test defined by the hypotheses

$$\mathcal{H}_0 : \bar{R}_t \leq R_0 + \epsilon_{\text{tol}}, \quad \forall t \geq 1, \tag{4a}$$

$$\mathcal{H}_1 : \exists t^* \geq 1 \text{ such that } \bar{R}_{t^*} > R_0 + \epsilon_{\text{tol}}. \tag{4b}$$

The null hypothesis $\mathcal{H}_0$ stipulates that the running test risk never exceeds the nominal risk $R_0$ by more than the tolerance $\epsilon_{\text{tol}}$, while the alternative hypothesis $\mathcal{H}_1$ covers cases in which the running risk exhibits a *harmful degradation*, i.e., $\bar{R}_{t^*} > R_0 + \epsilon_{\text{tol}}$ for at least one time $t^*$.

In practice, the nominal risks $R_t$, for all indices $t = 0, 1, ...$, are not directly accessible, but they can be estimated based on the available data via the *empirical risks*

$$\hat{R}_t = \frac{1}{n_t} \sum_{i=1}^{n_t} \ell(f(x_{t,i}), y_{t,i}) \tag{5}$$

for $t = 0, 1, ...$ Using these estimates, at each time step $t$, a sequential test processes the first $t$ estimates $\hat{R}_0, \hat{R}_1,..., \hat{R}_t$ to output a decision

$$\Phi_t = \begin{cases} 0, & \text{continue testing,} \\ 1, & \text{reject hypothesis } \mathcal{H}_0 \quad \text{(alarm).} \end{cases} \tag{6}$$

Accordingly, a decision $\Phi_t = 1$ indicates that there is enough evidence to reject the null hypothesis $\mathcal{H}_0$ that no

harmful shift has occurred, while the decision $\Phi_t = 0$ causes the test to continue operating.

The design goal for the decision rule is to control the type-I error, i.e., the probability of producing decision $\Phi_t = 1$ when the null hypothesis $\mathcal{H}_0$ of no harmful shifts holds true. The type-I error requirement can be formulated as the inequality

$$\mathbb{P}_{\mathcal{H}_0}\left(\exists t \geq 1 : \Phi_t = 1\right) \leq \delta, \tag{7}$$

where $\delta \in (0, 1)$ is a user-defined value, and $P_{\mathcal{H}_0}$ denotes any probability sequence $P_1, P_2, ...$ satisfying the condition (4a).

Besides guaranteeing the type-I error (7), one is also interested in reducing the average time required for correctly detecting a harmful shift. Formally, since the earliest time an alarm is produced as $\min\{t \geq 1 : \Phi_t = 1\}$, the average time to alarm under some sequence of distributions belonging to the alternative $\mathcal{H}_1$ is defined as the expected stopping time

$$T = \mathbb{E}_{\mathcal{H}_1}\left[\min\left\{t \geq 1 : \Phi_t = 1\right\}\right]. \tag{8}$$

### 2.2. Supervised Risk Monitoring

Given the estimated risks $\hat{R}_0, \hat{R}_1, \ldots, \hat{R}_t$ in (5), the SRM method introduced in (Podkopaev & Ramdas, 2022) constructs an upper bound $U_0$ on the source risk $R_0$ and a lower bound $L_t$ on the running test risk $\bar{R}_t$ as

$$U_0 = \hat{R}_0 + w_0, \quad \text{and} \quad L_t = \hat{\bar{R}}_t - w_t, \tag{9}$$

where $w_0$ and $w_t$ are finite-sample correction terms, and $\hat{\bar{R}}_t = t^{-1}\sum_{t'=1}^{t} \hat{R}_{t'}$ denotes the empirical estimates of running risk.

The correction term $w_0$ is selected so as to ensure that the true source risk $R_0$ is upper bounded by $U_0$ with probability no smaller than $1 - \delta_S$, i.e.,

$$\mathbb{P}\left(R_0 \leq U_0\right) \geq 1 - \delta_S, \tag{10}$$

for a given probability $\delta_S \in (0, 1)$. For example, the constant $w_0$ can be obtained using Hoeffding's inequality as $w_0 = \sqrt{\ln\left(1/\delta_S\right)/2n_0}$, given that the loss is assumed to be bounded (Hoeffding, 1963; Einbinder et al., 2025).

In contrast, the constants $w_t$ are chosen such that the sequence $\{L_t\}_{t\geq 1}$ forms an *anytime-valid* lower bound on the running test risk. Formally, this requirement is captured by the inequality

$$\mathbb{P}\big(\bar{R}_t \geq L_t, \forall t \geq 1\big) \geq 1 - \delta_T, \tag{11}$$

for some confidence level $\delta_T \in (0, 1)$. This bound can be obtained by leveraging the following lemma.

**Lemma 2.1** (Theorem 4, Conjugate-mixture empirical Bernstein (CM-EB) adapted from (Howard et al., 2021))**.**

*Consider a sequence of random variables $\{Z_t\}_{t \geq 1}$ with $Z_t \in [0, 1]$ for all $t \geq 1$. Define the time-averaged mean $\mu_t = t^{-1} \sum_{t'=1}^{t} \mathbb{E}[Z_{t'}]$, and the empirical mean $\hat{\mu}_t = t^{-1} \sum_{t'=1}^{t} Z_{t'}$. Let $\{\hat{Z}_t\}_{t \geq 1}$ be any predictable sequence with $\hat{Z}_t \in [0, 1]$, in the sense that each variable $\hat{Z}_t$ is a function of the past variables $Z_1, \ldots, Z_{t-1}$ only. Define the function $u(V_t) = \sup \{ S : \int_0^1 q(\lambda) \exp(\lambda S - \psi_E(\lambda) V_t) \, \mathrm{d}\lambda < 1/\delta_{\mathrm{T}} \}$ for any distribution $q(\lambda)$ on parameter $\lambda$, where the supremum is taken over the scalar $S$, $V_t = \sum_{t'=1}^{t} (Z_{t'} - \hat{Z}_{t'})^2$ is the cumulative prediction error, $\psi_E(\lambda) = -\log(1 - \lambda) - \lambda$, and $\delta_{\mathrm{T}} \in (0, 1)$ is a probability. Then, we have the inequality*

$$\mathbb{P}\left( \forall t \geq 1 : |\mu_t - \hat{\mu}_t| < \frac{u(V_t)}{t} \right) \geq 1 - 2\delta_{\mathrm{T}}. \quad (12)$$

In the context of SRM, the confidence sequence (12) in Lemma 2.1 can be leveraged to obtain an anytime-valid lower confidence bound (9) with the correction term $w_t = u(V_t)/t$, where $V_t$ becomes $V_t = \sum_{t'=1}^{t} (\hat{R}_{t'} - \hat{\hat{R}}_{t'-1})^2$.

By combining both bounds, i.e., (10) and (11), SRM obtains the decision variable as

$$\Phi_t = \mathbb{1}[L_t > U_0 + \epsilon_{\mathrm{tol}}], \quad (13)$$

where $\mathbb{1}[\cdot]$ is the indicator function. As shown in Figure 1, this decision rule triggers an alarm when the lower bound on the running test risk $\bar{R}_t$ exceeds the upper bound on source risk $R_0$ plus the tolerance margin $\epsilon_{\mathrm{tol}}$. Accordingly, the average earliest detection time (8) is given by

$$T = \mathbb{E}[\min\{t \geq 1 : L_t > U_0 + \epsilon_{\mathrm{tol}}\}]. \quad (14)$$

SRM controls the type-I error (7) as long as one chooses the probabilities in (10) and (11) as $\delta = \delta_{\mathrm{S}} + \delta_{\mathrm{T}}$, as formalized in the following lemma.

**Lemma 2.2** (Type-I Error Guarantee of SRM (Podkopaev & Ramdas, 2022)). *Given user-defined miscoverage levels $\delta_{\mathrm{S}}, \delta_{\mathrm{T}} \in (0, 1)$, satisfying the condition $\delta_{\mathrm{S}} + \delta_{\mathrm{T}} \in (0, 1)$, SRM provides the following guarantee on the type-I error:*

$$\mathbb{P}_{\mathcal{H}_0}(\exists t \geq 1, \Phi_t = 1) \leq \delta_{\mathrm{S}} + \delta_{\mathrm{T}}. \quad (15)$$

## 3. Semi-Supervised Risk Monitoring

In many real-world deployments, acquiring labeled feedback is costly and time-consuming, while unlabeled test data are abundant and arrive continuously. As shown in Figure 1, this motivates a semi-supervised paradigm where the risk monitor can leverage a small set of labeled samples, together with a larger pool of unlabeled observations. To address this setting, in this section, we propose a generalization of SRM (Podkopaev & Ramdas, 2022) referred to as *prediction-powered semi-supervised risk monitoring* (PPRM), which can leverage both labeled and unlabeled data.

### 3.1. Prediction-Powered Risk Estimates

As shown in Figure 1, at each time step $t = 1, 2, \ldots$, the monitor observes a labeled batch $\mathcal{D}_t = \{(x_{t,i}, y_{t,i})\}_{i=1}^{n_t}$, as for SRM, along with an unlabeled batch $\tilde{\mathcal{D}}_t = \{\tilde{x}_{t,i}\}_{i=n_t+1}^{n_t+N_t}$ of $N_t$ inputs with no associated labels. The main challenge in this setting lies in how to efficiently combine the labeled and unlabeled streams while controlling the type-I error (7). We emphasize that, unlike (Schirmer et al., 2025; Amoukou et al., 2024), we assume that the number of labeled data points, $n_t$, is strictly positive. As we will see, this allows us to avoid the technical assumptions required in (Schirmer et al., 2025) about the error rates of the models used to generate synthetic labels for the unlabeled data.

PPRM incorporates PPI-based estimates of the risks (Angelopoulos et al., 2023a) into the SRM framework. Through PPI, PPRM constructs unbiased empirical estimates of the risks by: (*i*) using a predictive model $f_{\mathrm{p}}(\cdot)$ to generate synthetic labels for the unlabeled portion of the data; and then (*ii*) correcting for the systematic bias caused by the synthetic labels via calibration using the labeled samples. The labeling model $f_{\mathrm{p}}(\cdot)$ may be, for instance, a general-purpose expert, such as an LLM, or possibly the deployed model $f(\cdot)$ itself. As anticipated, unlike (Schirmer et al., 2025), we do not make any assumptions on the quality of the model $f_{\mathrm{p}}(\cdot)$.

Write as $\tilde{y}_{t,i}$ the synthetic labels for the unlabeled input $\tilde{x}_{t,i}$, i.e.,

$$\tilde{y}_{t,i} = \arg\max_c f_{\mathrm{p}}(\tilde{x}_{t,i})[c], \quad (16)$$

where $f_{\mathrm{p}}(\cdot)[c]$ denotes the predicted probability assigned to class $c$ by the predictive model $f_{\mathrm{p}}(\cdot)$. Using PPI, the source risk $R_0$ is estimated as

$$\hat{R}_0^{\mathrm{PP}} = \underbrace{\frac{\eta_0}{N_0} \sum_{j=n_0+1}^{n_0+N_0} \ell(f(\tilde{x}_{0,j}), \tilde{y}_{0,j})}_{\hat{R}_0^{\mathrm{U}}}$$
$$+ \underbrace{\frac{1}{n_0} \sum_{i=1}^{n_0} \ell(f(x_{0,i}), y_{0,i}) - \frac{\eta_0}{n_0} \sum_{i=1}^{n_0} \ell(f(x_{0,i}), \tilde{y}_{0,i})}_{\hat{R}_0^{\mathrm{rect}}},$$
$$(17)$$

where $\eta_0 \geq 0$ is a hyperparameter that controls the reliance of the estimate on the unlabeled data (Angelopoulos et al., 2023b). The first term $\hat{R}_0^{\mathrm{U}}$ in (17) is the empirical risk evaluated only with the unlabeled synthetic data, while the second term $\hat{R}_0^{\mathrm{rect}}$ estimates the bias introduced by the imputation. As proved in (Angelopoulos et al., 2023a), using PPI++, the empirical estimate $\hat{R}_0^{\mathrm{PP}}$ is unbiased, in the sense that $\mathbb{E}[\hat{R}_0^{\mathrm{PP}}] = R_0$, for any choice of hyperparameter $\eta_0 \geq 0$.

In a similar way, the running test risk $\bar{R}_t$ is estimated as

$$
\hat{\bar{R}}_t^{\mathrm{PP}} = \frac{1}{t} \sum_{t'=1}^{t} \left[ \underbrace{\frac{\eta_{t'}}{N_{t'}} \sum_{j=n_{t'}+1}^{n_{t'}+N_{t'}} \ell(f(\tilde{x}_{t',j}), \tilde{y}_{t',j})}_{\hat{R}_{t'}^{\mathrm{U}}} \right.
$$
$$
\left. + \underbrace{\frac{1}{n_{t'}} \sum_{i=1}^{n_{t'}} \ell(f(x_{t',i}), y_{t',i}) - \frac{\eta_{t'}}{n_{t'}} \sum_{i=1}^{n_{t'}} \ell(f(x_{t',i}), \tilde{y}_{t',i})}_{\hat{R}_{t'}^{\mathrm{rect}}} \right],
$$
$$(18)$$

using both labeled and unlabeled data, where $\eta_t$ plays the same role as in (17). Furthermore, as in (17), the first term $\hat{R}_t^{\mathrm{U}}$ in (18) denotes the empirical risk evaluated only with the unlabeled synthetic data at time step $t$, while the second term $\hat{R}_t^{\mathrm{rect}}$ estimates the bias introduced by the imputation. We write

$$
\hat{R}_t^{\mathrm{PP}} = \hat{R}_t^{\mathrm{U}} + \hat{R}_t^{\mathrm{rect}} \tag{19}
$$

for the contribution to the estimate (18) corresponding to each time step $t$.

**Lemma 3.1** (Unbiasedness of PPRM). *Assume that the hyperparameters $\{\eta_t\}_{t\geq 1}$ form a predictable sequence, in the sense that each hyperparameter $\eta_t$ is a function only of the data $\{\mathcal{D}_{t'}, \tilde{\mathcal{D}}_{t'}, \{\tilde{y}_{t',i}\}_{i=1}^{n_{t'}+N_{t'}}\}_{t'=1}^{t-1}$ observed strictly before time $t$. Then, the quantity (18) is an unbiased estimate of the corresponding true risk, i.e.,*

$$
\mathbb{E}\left[\hat{\bar{R}}_t^{\mathrm{PP}}\right] = \bar{R}_t. \tag{20}
$$

The proof can be found in Appendix A.1.

## 3.2. Upper and Lower PPRM Bounds

As in SRM, PPRM obtains an upper bound $U_0^{\mathrm{PP}}$ on the source risk, as well as a lower confidence bound $L_t^{\mathrm{PP}}$ for $t = 1, 2, \ldots$, which are denoted respectively by

$$
U_0^{\mathrm{PP}} = \hat{R}_0^{\mathrm{PP}} + w_0^{\mathrm{PP}} \quad \text{and} \quad L_t^{\mathrm{PP}} = \hat{\bar{R}}_t^{\mathrm{PP}} - w_t^{\mathrm{PP}}. \tag{21}
$$

The correction term $w_0^{\mathrm{PP}}$ is designed to ensure the inequality $\mathbb{P}(R_0 \leq U_0^{\mathrm{PP}}) \geq 1 - \delta_{\mathrm{S}}$, and we adopt here the betting-based method from (Einbinder et al., 2025). Further details are provided in Appendix A.2. To compute the correction terms $w_t^{\mathrm{PP}}$, we apply the confidence sequence bound in Lemma 2.1 to the PPI-based risk estimates.

To this end, since the prediction-powered estimates $\hat{R}_t^{\mathrm{PP}}$ take values in the interval $[-\eta_t, \eta_t + 1]$, we first apply an affine normalization $g_a(\cdot)$, i.e., $g_a(\ell) = (\ell + \eta_{\max})/(1 + 2\eta_{\max})$, to the loss function $\ell$ so that each estimate $\hat{R}_t^{\mathrm{PP}}$ lies in the range $[0, 1]$, where $\eta_{\max}$ is the maximum permitted value for $\eta_t$. With the rescaled loss, the correction term $w_t^{\mathrm{PP}}$ is given by $w_t^{\mathrm{PP}} = u(V_t^{\mathrm{PP}})/t$, with $u(V_t^{\mathrm{PP}}) = \sup\{S : \int_0^1 q(\lambda) \exp(\lambda S - $

$\psi_E(\lambda) V_t^{\mathrm{PP}}) \, d\lambda < 1/\delta_{\mathrm{T}}\}$, where

$$
V_t^{\mathrm{PP}} = \sum_{t'=1}^{t} \left(\hat{R}_{t'}^{\mathrm{PP}} - \hat{\bar{R}}_{t'-1}^{\mathrm{PP}}\right)^2. \tag{22}
$$

The term $V_t^{\mathrm{PP}}$ in (22) is the cumulative squared prediction error for the estimate $\hat{R}_{t'}^{\mathrm{PP}}$ in (19) when using the corresponding time-average up to time $t'-1$, i.e., $\hat{\bar{R}}_{t'-1}^{\mathrm{PP}}$ in (18), as the prediction.

PPRM then applies the decision rule as

$$
\Phi_t^{\mathrm{PP}} = \mathbb{1}\left[L_t^{\mathrm{PP}} > U_0^{\mathrm{PP}} + \epsilon_{\mathrm{tol}}\right]. \tag{23}
$$

Following the same argument as Lemma 2.2, we have the following result.

**Theorem 3.2** (Type-I Error Guarantee of PPRM). *Given user-defined miscoverage levels $\delta_{\mathrm{S}}, \delta_{\mathrm{T}} \in (0, 1)$, satisfying the condition $\delta_{\mathrm{S}} + \delta_{\mathrm{T}} \in (0, 1)$, let $\{\eta_t\}_{t\geq 1}$ be any predictable sequence, as defined in Lemma 3.1. Then, PPRM provides the following guarantee on the type-I error:*

$$
\mathbb{P}_{\mathcal{H}_0}\left(\exists\, t \geq 1 : \Phi_t^{\mathrm{PP}} = 1\right) \leq \delta_{\mathrm{S}} + \delta_{\mathrm{T}}. \tag{24}
$$

To prove this result, it suffices to note that, as long as the sequence $\{\eta_t\}_{t\geq 1}$ is predictable, the prediction-powered risk estimates (17) and (18) are unbiased by Lemma 3.1. The proof then follows directly from Lemma 2.2.

## 3.3. Optimizing the Reliance on Unlabeled Data

The prediction-powered risk estimate $\hat{R}_t^{\mathrm{PP}}$ relies on the unlabeled data to an extent that depends on the hyperparameter $\eta_t$ by leveraging the result in Lemma 3.1, which allows $\eta_t$ to be a function of past data up to time $t-1$.

To this end, we propose to minimize the average value of the prediction error $V_t^{\mathrm{PP}}$ in (22), which determines the tightness of the lower bound $L_t^{\mathrm{PP}}$ in (21). A tighter bound ensures that, when the alternative hypothesis (4b) holds, the PPRM test (23) signals a harmful degradation, setting $\Phi_t^{\mathrm{PP}} = 1$, at an earlier time $t$.

**Lemma 3.3** (Optimal Choice of $\eta_t$ for Variance Reduction). *Let $\hat{R}_t^{\mathrm{PP}}(\eta_t)$ denote the prediction-powered empirical risk (19) at time $t$, which is parameterized by the hyperparameter $\eta_t$. The value of hyperparameter $\eta_t$ that minimizes the variance, i.e.,*

$$
\eta_t^* = \arg\min_{\eta_t \geq 0} \mathbb{E}\left[\left(\hat{R}_t^{\mathrm{PP}}(\eta_t) - \hat{\bar{R}}_{t-1}^{\mathrm{PP}}\right)^2\right], \tag{25}
$$

*is given by*

$$
\eta_t^* = \frac{\mathrm{Cov}(u_t, \tilde{u}_t^{\mathrm{L}})}{\left(1 + \frac{n_t}{N_t}\right) \mathrm{Var}(\tilde{u}_t^{\mathrm{U}})}, \tag{26}
$$

*where $u_t = \ell(f(x_t), y_t)$ denotes the true risk on a labeled sample; $\tilde{u}_t^{\mathrm{L}} = \ell(f(x_t), f_{\mathrm{p}}(x_t))$ and $\tilde{u}_t^{\mathrm{U}} = \ell(f(\tilde{x}_t), f_{\mathrm{p}}(\tilde{x}_t))$*

*denote the surrogate risk on a labeled sample and on an unlabeled sample, respectively; and $\mathrm{Var}(\cdot)$ and $\mathrm{Cov}(\cdot)$ represent variance and covariance.*

The proof of this result, which follows in a manner similar to (Angelopoulos et al., 2023b), can be found in Appendix A.3.

The hyperparameter $\eta_t^*$ in (26) depends on the true data distribution, which is unknown. Thus, we propose to estimate it using the data up to time step $t-1$ based on a sliding-window approach with fixed window size $L$. This produces a predictable sequence $\{\eta_t\}_{t \geq 1}$, satisfying the assumption of Theorem 3.2. Specifically, we consider the following plug-in estimator

$$\eta_t = \frac{\widehat{\mathrm{Cov}}_{\mathcal{H}_{t-1}}(u, \tilde{u})}{\left(1 + \frac{|\mathcal{H}_{t-1}|}{|\tilde{\mathcal{H}}_{t-1}|}\right) \widehat{\mathrm{Var}}_{\tilde{\mathcal{H}}_{t-1}}(\tilde{u})}, \qquad (27)$$

where $\widehat{\mathrm{Cov}}_{\mathcal{H}_{t-1}}(u, \tilde{u})$ and $\widehat{\mathrm{Var}}_{\tilde{\mathcal{H}}_{t-1}}(\tilde{u})$ denote the empirical estimates using the data

$$\mathcal{H}_{t-1} = \bigcup_{t'=t-L}^{t-1} \mathcal{D}_{t'}, \quad \text{and} \quad \tilde{\mathcal{H}}_{t-1} = \bigcup_{t'=t-L}^{t-1} \tilde{\mathcal{D}}_{t'}, \quad (28)$$

with $|\mathcal{H}_{t-1}|$ and $|\tilde{\mathcal{H}}_{t-1}|$ denoting the total numbers of historical labeled and unlabeled samples available before time $t$, respectively.

We further provide a theoretical analysis of the detection delay, under a simplified stationary regime with binary risks in Appendix A.4.

## 4. Related Works

**Sequential Testing-Based Risk Monitoring:** SRM was introduced by (Podkopaev & Ramdas, 2022) to tackle the problem of detecting harmful distribution shifts in a sequential setting. SRM builds confidence bounds for model risk using labeled calibration data (Podkopaev & Ramdas, 2022) and incoming test data, and provides time-uniform guarantees on type-I error control. Subsequently, Amoukou et al. extended this line of work to the unlabeled production-data setting, while Schirmer et al. further explored scenarios involving test-time adaptation models without access to labels. Related approaches by (Bar et al., 2024) and (Vovk et al., 2021) focus on identifying changes in certain statistical properties. However, such changes do not necessarily indicate harmful deviations.

Beyond tracking running risk, several studies have examined the instantaneous true risk with similar type-I error guarantees (Xu et al., 2024; Zecchin et al., 2025; Timans et al., 2025; Shekhar & Ramdas, 2023; Vovk & Wang, 2021). These methods are built around the testing-by-betting frame-

work (Ramdas & Wang, 2025). However, we target persistent performance degradation, while these methods are designed to detect instantaneous violations.

**Prediction-Powered Inference:** In many practical settings, models can access large pools of unlabeled data but only a limited amount of costly ground-truth labels. PPI was introduced to make use of predictions from an auxiliary model to facilitate statistical inference. Its key idea is to construct prediction-powered estimators that remain unbiased for the target risk, while using predictive information to reduce variance. The framework has since been extended in several directions, including robust PPI++ (Angelopoulos et al., 2023b), local PPI (Gu & Xia, 2024), anytime-valid PPI (Kilian et al., 2025), active PPI (Zrnic & Candès, 2024), and FAB-PPI (Cortinovis & Caron, 2025). More recently, Csillag et al. introduced prediction-powered e-values (Shafer & Vovk, 2019) for sequential change-point detection (Shekhar & Ramdas, 2023; Shin et al., 2022). Their approach also builds on the testing-by-betting paradigm (Ramdas & Wang, 2025), and therefore differs from our proposed PPRM both in the hypothesis being tested and in the experimental setup.

**LLM Performance Evaluation:** Selecting an appropriate model from a number of candidates requires reliable estimates of each model's performance. Traditional evaluation methods depend on deploying models and gathering empirical evidence through real-world testing, which is often expensive to deploy. To mitigate this burden, recent studies have investigated autoevaluation techniques that automate assessment using synthetic data or automated tools, thereby reducing the need for human involvement (Chaganty et al., 2018; Zheng et al., 2023). PPI has also been applied to improve the efficiency of model evaluation (Fisch et al., 2024; Park et al., 2025; Einbinder et al., 2025). Nonetheless, the problem of monitoring a deployed model using PPI has not been explored.

## 5. Experiments

In this section, we present two case studies, including image classification and LLM monitoring, and defer additional results including for a telecommunications task, to the appendix.

**Baselines** We compare PPRM to several baseline monitors:

- **Ideal PPRM**: This is an idealized case where PPRM has full access to the true labels of both labeled dataset $\mathcal{D}_t$ and unlabeled dataset $\tilde{\mathcal{D}}_t$. This produces the performance upper bound corresponding to full label access.
- **SRM** (Podkopaev & Ramdas, 2022): As discussed in Section 2.2, SRM uses only the labeled data $\mathcal{D}_t$.

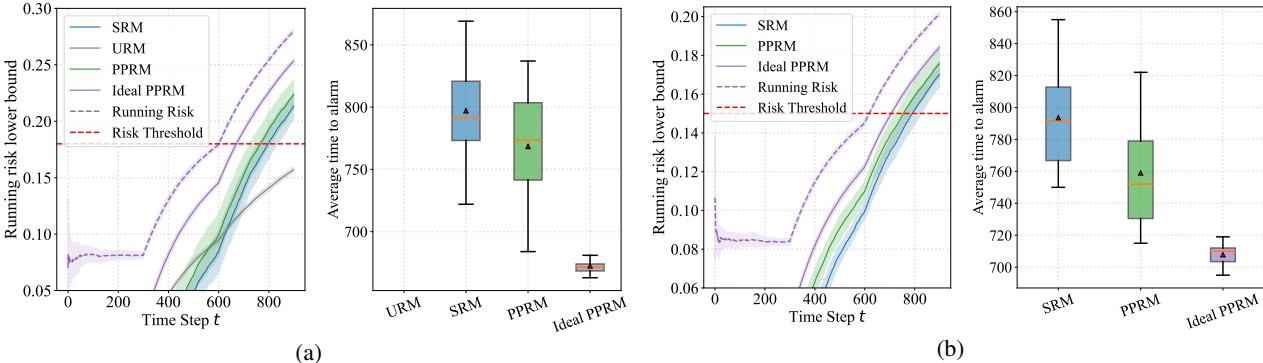

(a)                                                                                    (b)

*Figure 2.* Risk estimates as a function of time $t$ and average time to alarm for an image classification task under increasing shift severity: (a) binary loss monitored with an external predictor; (b) squared loss monitored using labels produced by the deployed model itself.

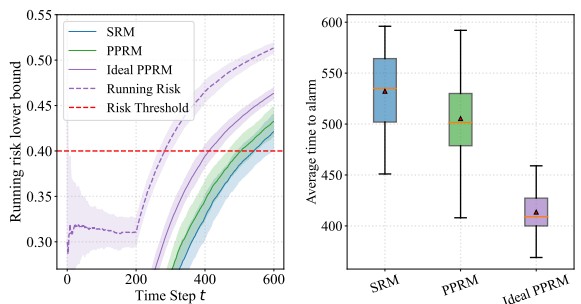

*Figure 3.* Performance for the LLM QA task under prompt shifts introduced at time $t = 200$.

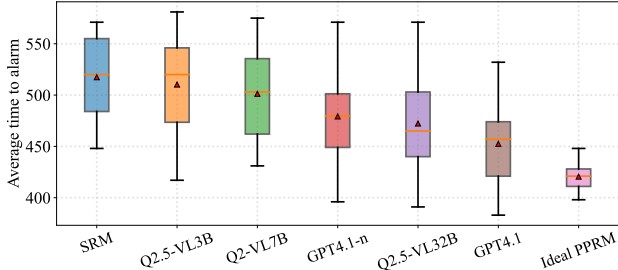

*Figure 4.* Average time to alarm for the LLM QA task using different predictors $f_p(\cdot)$ to produce synthetic labels.

- **Unsupervised risk monitoring (URM)** (Schirmer et al., 2025): This scheme uses only unlabeled data and is reviewed in Appendix B.

We focus on binary 0–1 loss and squared loss. Unless otherwise specified, we set $\delta = \delta_S + \delta_T = 0.25$, allocating most of the budget to controlling the test risk, i.e., $\delta_T = 0.2$ and $\delta_S = 0.05$. We set $\eta_0 = 1$ in (17), and the window length $L$ to obtain $\eta_t$ in (27) is set to 60, numerical validations of these choices can be found in Appendix D.1. The numbers of labeled data samples and unlabeled data samples at each time step are set to 1 and 15. To compute all the lower bounds of the running risk, we adopt the CM-EB method reviewed in Lemma 2.1. For all figures, we show confidence intervals constructed spanning one standard deviation of the empirical distribution across all runs on both sides of the mean.

### 5.1. Sequential Risk Monitoring on Image Data

#### 5.1.1. PROBLEM FORMULATION

We first consider the task of risk monitoring in the vision domain, replicating the setting of (Schirmer et al., 2025), where the CIFAR10-C (Hendrycks & Dietterich, 2019)

dataset was considered. Accordingly, the test-time distribution is obtained by applying Gaussian noise with three levels of severity to the image distribution corresponding to the nominal condition. We monitor the performance of a ResNet-32 model (He et al., 2016) trained on nominal data. To generate synthetic labels for unlabeled samples, we employ the ResNet-1201 model trained on the same dataset.

#### 5.1.2. RESULTS

**Monitoring with Drifting Data Distribution:** To simulate an increasing test risk, we gradually increase the severity level of Gaussian noise corruption from nominal data, i.e., severity level 0, up to severity level 2. Figure 2(a) shows the results of the estimated test risk and of the alarm decisions. As the corruption intensifies, the true risk exceeds the safe threshold at time $t = 600$, PPRM raises an alarm at around time $t = 760$, whereas SRM reacts more slowly, at around time $t = 800$ due to high estimation variance, and URM fails to trigger an alarm within the considered time period.

**Monitoring with Self-Synthetic Strategy:** The proposed PPRM can also be applied when the model itself is used to provide label surrogates for the unlabeled data. For this setting, focusing on $C$-class classification, we consider the

squared loss between labels and model confidence levels, i.e., $\ell(f(x), y) = \frac{1}{2} \sum_{c=1}^{C} (f(x)_c - \mathbb{1}[y = c])^2$, as the loss function. This squared loss corresponds to the Brier score when averaged across samples, and it was also adopted in (Schirmer et al., 2025).

As seen in Figure 2(b), even without an additional predictor, PPRM can reduce the time to alarm by making use of the model's intrinsic confidence signals. In particular, an alarm is raised at around time $t = 760$, while SRM requires $t = 790$ time steps.

## 5.2. Monitoring an LLM with Limited Human-Labeled Data

### 5.2.1. PROBLEM FORMULATION

We now focus on an LLM deployed to address *question-answering* (QA) tasks. We draw from several established benchmarks, including MMLU (Hendrycks et al., 2021), CMExam (Liu et al., 2023), CommonsenseQA (Talmor et al., 2019), Social-IQA, and PubMedQA (Jin et al., 2019). We use Qwen2-VL-2B (Wang et al., 2024) as the deployed model whose performance we aim to monitor. For the monitoring LLM, we experiment with a diverse set of both open-source and closed-source models.

### 5.2.2. RESULTS

To simulate an environment in which risk increases over time, starting from time $t = 200$, we introduce more complex prompts that the deployed model cannot answer correctly, causing the model's performance to deteriorate. These complex prompts are obtained by retaining cases where a different model, namely Qwen2.5-VL-7B, fails to produce correct answers.

**Monitoring with Drifting Data Distribution:** As shown in Figure 3, confirming the results in the previous example, PPRM significantly reduces the average time to alarm. In particular, while the true risk exceeds the allowed level at $t = 280$, PPRM raises an alarm at around time $t = 500$, whereas SRM raises an alarm at around time $t = 530$.

We then further assess how the accuracy of the LLM predictor affects performance by testing a range of different labeling models, namely Qwen2.5-VL-3B (abbreviated as "Q2.5-VL3B"), Qwen2.5-VL-32B (abbreviated as "Q2.5-VL32B") (Bai et al., 2025), Qwen2-VL-7B (abbreviated as "Q2-VL7B") (Wang et al., 2024), GPT4.1-nano (abbreviated as "GPT4.1-n"), and GPT4.1 (Achiam et al., 2023).

The results are shown in Figure 4 by ordering the models from least to most powerful. As seen in the figure, when the LLM predictor exhibits higher accuracy, i.e., for larger models, the estimated accuracy of synthetic labels increases, leading to a tighter bound on the monitored model's running

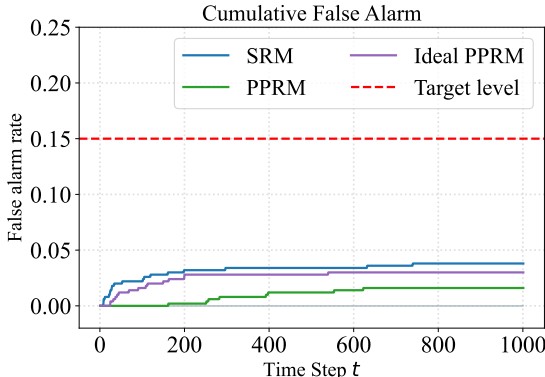

*Figure 5.* False-alarm rate as a function of time $t$ for the LLM QA task. The dashed line indicates the target false-alarm rate $\delta_T = 0.15$.

*Table 1.* Average time to alarm for PPRM with a fixed or an adaptive hyperparameter $\eta_t$.

| Predictor $f_p(\cdot)$ | PPRM (Fixed) | PPRM (Adaptive) |
|---|---|---|
| Q2.5-VL3B | 527.80 | 508.83 |
| Q2-VL7B | 519.08 | 498.31 |
| GPT4.1-n | 491.80 | 476.98 |
| Q2.5-VL32B | 486.22 | 471.85 |
| GPT4.1 | 460.69 | 449.88 |

risk, and consequently to a lower average time to alarm.

In Figure 5, we report the false-alarm rate as a function of time $t$ for the LLM QA task. We run multiple independent trials under the null setting, and record the entire trajectory of the lower confidence bound over time. For each trial, we check at every time step whether the bound has crossed the risk threshold, and convert this trajectory into a cumulative indicator, which equals one once a crossing has occurred at any previous time. The reported curve is obtained by averaging these cumulative indicators across trials at each time step. The results illustrate that the false-alarm rate remains below the target level across all time steps, thereby validating the theoretical guarantees of PPRM.

**Effect of Hyperparameter Optimization:** Table 1 reports the average time to alarm for PPRM with a fixed hyperparameter $\eta_t = 1$ and with an adaptive hyperparameter $\eta_t$ in (27). The two PPRM variants are identified as "PPRM (Fixed)" and "PPRM (Adaptive)", respectively. We consider the different imputation predictors $f_p(\cdot)$ considered in Figure 4. The table illustrates the benefits of the adaptive selection, particularly for small- and medium-sized predictors. For example, for the Qwen2-VL-7B model, the average time to alarm decreases from around 519 to around 498 thanks to adaptive hyperparameter selection. It is also observed that, as the predictor becomes stronger, the performance gap

between PPRM with fixed and adaptive hyperparameters gradually narrows. This demonstrates that adaptive hyperparameter selection, determining the reliance on unlabeled data, is particularly important for less effective imputation models.

## 6. Conclusion

In this work, we have presented a novel framework for sequential semi-supervised risk monitoring, extending classical supervised approaches to leverage both labeled and unlabeled data. By integrating PPI, our method, termed PPRM, provided unbiased estimates of the running test risk, while maintaining rigorous statistical guarantees on the type-I error. Through extensive experiments on three tasks, we have demonstrated that the proposed approach consistently achieved timely detection of harmful distribution shifts and outperformed purely supervised or unsupervised baselines. Future work may consider methods that integrate PPRM with unsupervised techniques such as test-time adaptation. Other interesting research directions include applications to engineering problems, e.g., in robotics.

## Acknowledgements

The work of Guangyi Zhang and Yunlong Cai was supported in part by the National Natural Science Foundation of China under Grant 62571477, and in part by Zhejiang Provincial Key Laboratory of Multi-Modal Communication Networks and Intelligent Information Processing, Hangzhou 310027, China. The work of Osvaldo Simeone was supported by the European Research Council (ERC) under the European Union's Horizon Europe Programme (grant agreement No. 101198347), by an Open Fellowship of the EPSRC (EP/W024101/1), and by the EPSRC project (EP/X011852/1).

## Impact Statement

This paper presents work whose goal is to advance the field of machine learning. The impact of this paper may be more pronounced in engineering applications, such as telecommunication and robotics.

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

# A. Proofs and Analysis

## A.1. Proof of Lemma 3.1

We aim to show the equality $\mathbb{E}[\hat{\bar{R}}_t^{\text{PP}}] = \bar{R}_t = t^{-1} \sum_{t'=1}^{t} R_{t'}$, where $R_{t'} = \mathbb{E}[\ell(f(x_{t'}), y_{t'})]$. Recall that we have

$$
\begin{aligned}
\hat{\bar{R}}_t^{\text{PP}} &= \frac{1}{t} \sum_{t'=1}^{t} \left[ \hat{R}_{t'}^{\text{U}} + \hat{R}_{t'}^{\text{rect}} \right] \\
&= \frac{1}{t} \sum_{t'=1}^{t} \left[ \frac{\eta_{t'}}{N_{t'}} \sum_{j=n_{t'}+1}^{n_{t'}+N_{t'}} \ell\left(f(\tilde{x}_{t',j}), \tilde{y}_{t',j}\right) + \frac{1}{n_{t'}} \sum_{i=1}^{n_{t'}} \ell\left(f(x_{t',i}), y_{t',i}\right) - \frac{\eta_{t'}}{n_{t'}} \sum_{i=1}^{n_{t'}} \ell\left(f(x_{t',i}), \tilde{y}_{t',i}\right) \right].
\end{aligned}
\tag{29}
$$

By linearity of expectation, we have

$$
\mathbb{E}\left[ \hat{\bar{R}}_t^{\text{PP}} \right] = \frac{1}{t} \sum_{t'=1}^{t} \mathbb{E}\left[ \hat{R}_{t'}^{\text{U}} + \hat{R}_{t'}^{\text{rect}} \right].
\tag{30}
$$

For each term, we further have

$$
\mathbb{E}[\hat{R}_t^{\text{U}} + \hat{R}_t^{\text{rect}}] = \mathbb{E}_{\eta_t}[\mathbb{E}[\hat{R}_t^{\text{U}} + \hat{R}_t^{\text{rect}} | \eta_t]].
\tag{31}
$$

Since the hyperparameter $\eta_t$ is only a function of data observed at times $t' = 1, ..., t-1$, it is independent of the data observed at time $t$, and we have

$$
\begin{aligned}
\mathbb{E}[\hat{R}_t^{\text{U}} + \hat{R}_t^{\text{rect}} | \eta_t] &= \eta_t \cdot \mathbb{E}[\ell(f(\tilde{x}_t), \tilde{y}_t)] + \mathbb{E}[\ell(f(x_t), y_t) - \eta_t \cdot \ell(f(x_t), \tilde{y}_t)] \\
&= \mathbb{E}[\ell(f(x_t), y_t)] = R_t.
\end{aligned}
\tag{32}
$$

Then, this yields $\mathbb{E}[\hat{R}_t^{\text{U}} + \hat{R}_t^{\text{rect}}] = \mathbb{E}_{\eta_t}[R_t] = R_t$. Therefore, the equality $\mathbb{E}[\hat{\bar{R}}_t^{\text{PP}}] = t^{-1} \sum_{t'=1}^{t} R_{t'} = \bar{R}_t$ holds. This completes the proof.

## A.2. Source Upper Bound

The prediction-powered risk estimator on the source domain is given by (17), i.e.,

$$
\hat{R}_0^{\text{PP}} = \underbrace{\frac{\eta_0}{N_0} \sum_{j=n_0+1}^{n_0+N_0} \ell\left(f(\tilde{x}_{0,j}), \tilde{y}_{0,j}\right)}_{\hat{R}_0^{\text{U}}} + \underbrace{\frac{1}{n_0} \sum_{i=1}^{n_0} \ell\left(f(x_{0,i}), y_{0,i}\right) - \frac{\eta_0}{n_0} \sum_{i=1}^{n_0} \ell\left(f(x_{0,i}), \tilde{y}_{0,i}\right)}_{\hat{R}_0^{\text{rect}}}.
\tag{33}
$$

We partition the unlabeled set into $n_0$ disjoint blocks, each containing $N_0/n_0$ points. The $i$-th labeled sample is paired with the unlabeled block $j \in [(i-1)N_0/n_0 + n_0 + 1, \; iN_0/n_0 + n_0]$. The block-wise estimator can be denoted as

$$
\hat{R}_0^{\text{PP}} = \frac{1}{n_0} \sum_{i=1}^{n_0} \left( \frac{n_0}{N_0} \sum_{j=(i-1)\frac{N_0}{n_0}+n_0+1}^{i\frac{N_0}{n_0}+n_0} \eta_0 \cdot \ell(f(\tilde{x}_{0,j}), \tilde{y}_{0,j}) + \ell(f(x_{0,i}), y_{0,i}) - \eta_0 \cdot \ell(f(x_{0,i}), \tilde{y}_{0,i}) \right).
\tag{34}
$$

For convenience, define the per-sample risk

$$
z_{0,i}^{\text{PP}} = \frac{n_0}{N_0} \sum_{j=(i-1)\frac{N_0}{n_0}+n_0+1}^{i\frac{N_0}{n_0}+n_0} \eta_0 \cdot \ell(f(\tilde{x}_{0,j}), \tilde{y}_{0,j}) + \ell(f(x_{0,i}), y_{0,i}) - \eta_0 \cdot \ell(f(x_{0,i}), \tilde{y}_{0,i}).
\tag{35}
$$

Thus, the prediction-powered estimate (17) is given by the average $\hat{R}_0^{\text{PP}} = n_0^{-1} \sum_{i=1}^{n_0} z_{0,i}^{\text{PP}}$. Therefore, the upper bound can be obtained by the algorithms proposed in (Waudby-Smith & Ramdas, 2024). By viewing $z_{0,i}^{\text{PP}}$ as new data points, we can naturally call the Betting algorithm to get the required upper bound of source risk. Since $R_0^{\text{PP}}$ is an unbiased estimate of $R_0$, the resulting bound is naturally valid.

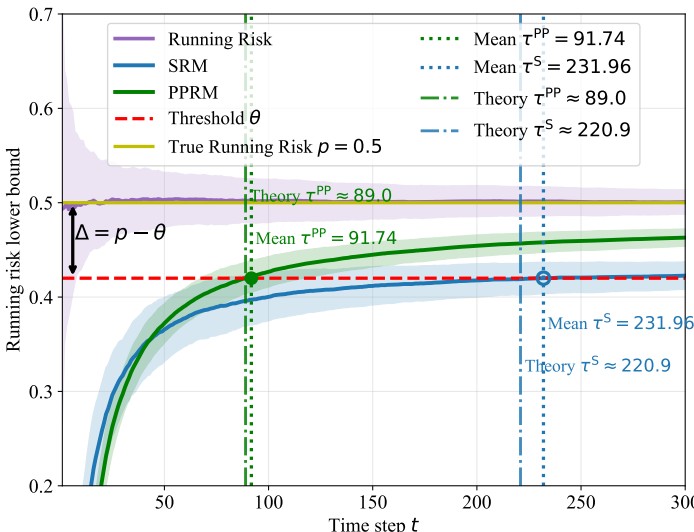

*Figure 6.* Detection performance comparison between SRM and PPRM under a stationary regime. The true risk is $p = 0.5$ and the threshold is $\theta = 0.42$. The dashed vertical lines indicate the time to alarm, while the solid vertical lines indicate the theoretical detection time computed using the approximations in (42).

## A.3. Proof of Lemma 3.3

For a fixed $\hat{\bar{R}}^{\mathrm{PP}}_{t-1}$, we have

$$\mathbb{E}\big[\big(\hat{R}^{\mathrm{PP}}_t(\eta_t) - \hat{\bar{R}}^{\mathrm{PP}}_{t-1}\big)^2\big] = \mathrm{Var}\big(\hat{R}^{\mathrm{PP}}_t(\eta_t)\big) + \big(\mathbb{E}[\hat{R}^{\mathrm{PP}}_t(\eta_t)] - \hat{\bar{R}}^{\mathrm{PP}}_{t-1}\big)^2. \tag{36}$$

Using $\mathbb{E}[\hat{R}^{\mathrm{PP}}_t(\eta_t)] = R_t$, this becomes

$$\mathbb{E}\big[\big(\hat{R}^{\mathrm{PP}}_t(\eta_t) - \hat{\bar{R}}^{\mathrm{PP}}_{t-1}\big)^2\big] = \mathrm{Var}\big(\hat{R}^{\mathrm{PP}}_t(\eta_t)\big) + \big(R_t - \hat{\bar{R}}^{\mathrm{PP}}_{t-1}\big)^2. \tag{37}$$

The second term depends only on $R_t$ and $\hat{\bar{R}}^{\mathrm{PP}}_{t-1}$, and is therefore independent of the hyperparameter $\eta_t$. Consequently, minimizing the expectation is equivalent to minimizing the variance term $\mathrm{Var}(\hat{R}^{\mathrm{PP}}_t(\eta_t))$. Noting that this variance minimization problem is the same as Example 6.1 of PPI++ (Angelopoulos et al., 2023b), we thus directly have the following optimal solution:

$$\eta^*_t = \frac{\mathrm{Cov}(u_t, \tilde{u}^{\mathrm{L}}_t)}{(1 + \frac{n_t}{N_t})\,\mathrm{Var}(\tilde{u}^{\mathrm{U}}_t)}, \tag{38}$$

## A.4. Theoretical Detection Time Comparison

We next provide a theoretical analysis to illustrate when PPRM can lead to shorter detection delay. For clarity, as shown in Figure 6, we consider a stationary regime with binary risks, where the system is already in the harmful regime when monitoring begins, so that the distribution shift is present from $t = 1$ onward.

Let $u_t = \ell(f(x_t), y_t)$ denote the true risk on a labeled sample; $\tilde{u}^{\mathrm{L}}_t = \ell(f(x_t), f_p(x_t))$ and $\tilde{u}^{\mathrm{U}}_t = \ell(f(\tilde{x}_t), f_p(\tilde{x}_t))$ denote the surrogate risk on a labeled sample and on an unlabeled sample, respectively. We assume $u^{\mathrm{U}}_t \sim \mathrm{Bernoulli}(q)$ and $u^{\mathrm{L}}_t \sim \mathrm{Bernoulli}(q)$. Let $u_t \sim \mathrm{Bernoulli}(p)$, $p > \theta$, with $\theta$ denoting the alarm threshold, and $\Delta = p - \theta > 0$ the risk margin. Let $\gamma = \mathrm{Cov}(u_t, \tilde{u}^{\mathrm{L}}_t)$. We further assume fixed batch sizes $n_t = n$ and $N_t = N$, and use a fixed reliance parameter $\eta_t = \eta$.

For SRM, the estimate $\hat{R}_t$ is the empirical mean of $n$ labeled samples, so its variance is

$$v^{\mathrm{S}} = \frac{p(1-p)}{n}. \tag{39}$$

For PPRM, the prediction-powered estimate $\hat{R}_t^{\text{PP}}(\eta)$ has variance

$$v^{\text{PP}}(\eta) = \eta^2 \frac{q(1-q)}{N} + \frac{1}{n}\left(p(1-p) + \eta^2 q(1-q) - 2\eta\gamma\right). \tag{40}$$

We consider the linear boundary associated with the sub-exponential transform $\psi_E(\lambda) = -\log(1-\lambda) - \lambda, \quad \lambda \in (0,1)$ ((Howard et al., 2020), Theorem 1)

$$u(v) = \frac{\log(1/\delta_{\text{T}})}{\lambda} + \frac{\psi_E(\lambda)}{\lambda}v. \tag{41}$$

In the considered scenario, $\hat{\bar{R}}_t$ converges almost surely to $p$. Moreover, since the variance process $V_t$ is formed from squared deviations around a predictable running mean that also converges to $p$, we use the large-$t$ approximations

$$\hat{\bar{R}}_t \approx p, \qquad V_t \approx tv^{\text{S}}. \tag{42}$$

For SRM, the lower confidence bound crosses the threshold when

$$p - \frac{u(V_t)}{t} > \theta. \tag{43}$$

Substituting the linear boundary and solving for $t$ gives the resulting detection delay

$$\tau^{\text{S}}(\lambda) = \frac{\log(1/\delta_{\text{T}})}{\lambda\Delta - \psi_E(\lambda)v^{\text{S}}}. \tag{44}$$

For PPRM, the prediction-powered estimate $\hat{R}_t^{\text{PP}}(\eta)$ lies in $[-\eta, 1+\eta]$. Therefore, before applying the linear boundary (41), we affinely normalize it to $[0,1]$. Let

$$\hat{X}_t^{\text{PP}} = \frac{\hat{R}_t^{\text{PP}}(\eta) + \eta}{1 + 2\eta} \in [0,1]. \tag{45}$$

On this normalized scale, the mean, threshold, and risk margin are respectively given by

$$\mathbb{E}[\hat{X}_t^{\text{PP}}] = \frac{p+\eta}{1+2\eta}, \qquad \theta_X = \frac{\theta+\eta}{1+2\eta}, \qquad \Delta_X = \mathbb{E}[\hat{X}_t^{\text{PP}}] - \theta_X = \frac{\Delta}{1+2\eta}. \tag{46}$$

The corresponding variance is

$$v_X^{\text{PP}}(\eta) = \frac{v^{\text{PP}}(\eta)}{(1+2\eta)^2}. \tag{47}$$

Thus, under the same approximation as (42), we have

$$\hat{\bar{X}}_t^{\text{PP}} \approx \frac{p+\eta}{1+2\eta}, \qquad V_{t,X}^{\text{PP}} \approx t\frac{v^{\text{PP}}(\eta)}{(1+2\eta)^2}. \tag{48}$$

The alarm condition for PPRM should therefore be written on the normalized scale as

$$\hat{\bar{X}}_t^{\text{PP}} - \frac{u(V_{t,X}^{\text{PP}})}{t} > \theta_X. \tag{49}$$

Solving for $t$ gives the approximate delay

$$\tau^{\text{PP}}(\eta) = \frac{\log(1/\delta_{\text{T}})}{\lambda\Delta/(1+2\eta) - \psi_E(\lambda)v^{\text{PP}}(\eta)/(1+2\eta)^2}. \tag{50}$$

Our numerical results in Figure 6 show that the derived approximate detection delay is close to the empirical detection delay observed in simulations. We can now state the resulting conclusion.

**Proposition A.1** (Approximate delay improvement condition). *Consider the above stationary regime, where distribution shift is present from $t = 1$ onward. Let the true labeled risk be binary with $u_t \sim \mathrm{Bernoulli}(p)$ and $p > \theta$, and let the predicted risk satisfy $\tilde{u}_t^L \sim \mathrm{Bernoulli}(q)$ with $\gamma = \mathrm{Cov}(u_t, \tilde{u}_t^L)$. Assume that the unlabeled predicted risk $\tilde{u}_t^U$ has the same marginal distribution as $\tilde{u}_t^L$. Define the margin $\Delta = p - \theta > 0$, fix the reliance parameter $\eta_t = \eta \geq 0$, and consider the linear boundary (([Howard et al., 2020](#)), Theorem 1) induced by $\psi_E(\lambda) = -\log(1 - \lambda) - \lambda$ for $\lambda \in (0, 1)$. Let $v^S$ and $v^{PP}(\eta)$ denote the variances of the SRM and PPRM estimators, respectively. Then the approximate deterministic crossing time of PPRM is smaller than that of SRM, i.e.,*

$$\tau^{PP}(\eta, \lambda) < \tau^S(\lambda), \tag{51}$$

*if and only if*

$$2\eta(1 + 2\eta)\lambda\Delta < \psi_E(\lambda)\left((1 + 2\eta)^2 v^S - v^{PP}(\eta)\right). \tag{52}$$

Specifically, an informative predictor can reduce the effective variance $v^{PP}(\eta)$ through its positive covariance with the true loss, thereby tightening the lower confidence sequence and accelerating detection. However, increasing $\eta$ also enlarges the range of the prediction-powered estimate from $[0, 1]$ to $[-\eta, 1 + \eta]$, which incurs the normalization penalty $(1 + 2\eta)^2$. Thus, PPRM improves the approximate detection delay when the induced variance reduction is large enough to offset this range-normalization penalty.

## B. Unsupervised Lower Confidence Sequences

Unsupervised risk monitoring has been recently proposed as a technique to track the performance of methods in scenarios where only an *unlabeled* test stream $\hat{\mathcal{D}}_t = \{\tilde{x}_{t,i}\}_{i=1}^{N_t}$ is available. To address the lack of labels, the authors of ([Schirmer et al., 2025](#)) replace a sequence of supervised losses with a sequence of loss proxies that can be computed directly from the unlabeled test stream. This approach enables the derivation of an unsupervised lower bound on the running test risk, which can then be used to design an unsupervised alarm function.

The core idea is to employ a proxy function $g(\cdot)$, serving as a loss proxy for the model $f(\cdot)$, defined as $r_t = g(x_t, f)$. In addition to being unsupervised (i.e., depending only on inputs $x_t$), the proxy should be at least partially informative about the corresponding loss variable. When $g(\cdot)$ satisfies the following assumption from ([Schirmer et al., 2025](#)), the running test risk can be lower bounded as follows.

**Assumption B.1** (Assumption 1, ([Schirmer et al., 2025](#))). *Given a sequence of losses $u_{0:t}$, let the corresponding sequence of loss proxies $r_{0:t}$ and proxy thresholds $\beta_0, \ldots, \beta_t \in \mathbb{R}$, along with a loss threshold $\tau$, be such that for all $t \geq 1$, the following inequality holds:*

$$\frac{1}{t}\sum_{t'=1}^{t}\underbrace{\mathbb{P}_{P_{t'}}\left(r_{t'} > \beta_{t'}, r_{t'} \leq \tau\right)}_{\mathrm{PFP}_{t'}} \leq \underbrace{\mathbb{P}_{P_0}\left(r_0 > \beta_0, u_0 \leq \tau\right)}_{\mathrm{PFP}_0} + \frac{1}{t}\sum_{t'=1}^{t}\underbrace{\mathbb{P}_{P_{t'}}\left(r_{t'} \leq \beta_{t'}, u_{t'} > \tau\right)}_{\mathrm{PFN}_{t'}}. \tag{53}$$

The running test risk can be lower bounded as

$$\bar{R}_t \geq \tau\left(\frac{1}{t}\sum_{t'=1}^{t}\mathbb{P}_{P_{t'}}(r_{t'} > \beta_{t'}) - \mathbb{P}_{P_0}(r_0 > \beta_0, u_0 \leq \tau)\right), \quad \forall t \geq 1. \tag{54}$$

Specifically, the authors of ([Schirmer et al., 2025](#)) proposed online threshold calibration to select $\beta_0, \ldots, \beta_t \in \mathbb{R}$ and $\tau$ by maximizing the F1 score based on the source model's proxy. Importantly, this bound depends only on the test loss proxies and the source loss, which means that its corresponding lower-bound confidence sequence can be evaluated using a combination of unlabeled test data $\{\hat{\mathcal{D}}_{t'}\}_{t'=1}^{t} = \{\{x_{t',i}\}_{i=1}^{N_{t'}}\}_{t'=1}^{t}$ and labeled source data $\mathcal{D}_0$. In practice, this requires computing an empirical estimate of $\mathbb{P}_{P_{t'}}(r_{t'} > \beta_{t'})$ at each step, followed by the construction of a lower confidence sequence relying on specific bounds. This makes it suitable for the final proposed unsupervised alarm:

$$\Phi_t^b = \mathbb{1}\left[L_t\left(r_{0:t}, \beta_{0:t}, u_0, \tau\right) > U_0(r_0) + \epsilon_{\mathrm{tol}}\right]. \tag{55}$$

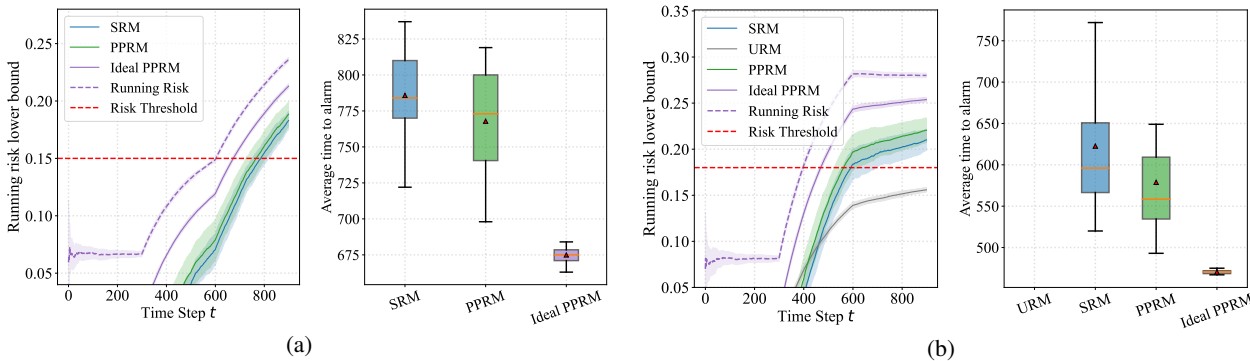

*Figure 7.* Risk estimates as a function of time $t$ and average time to alarm for an image classification task: (a) under increasing shift severity (squared loss); (b) under a non-monotonic shift severity (binary loss).

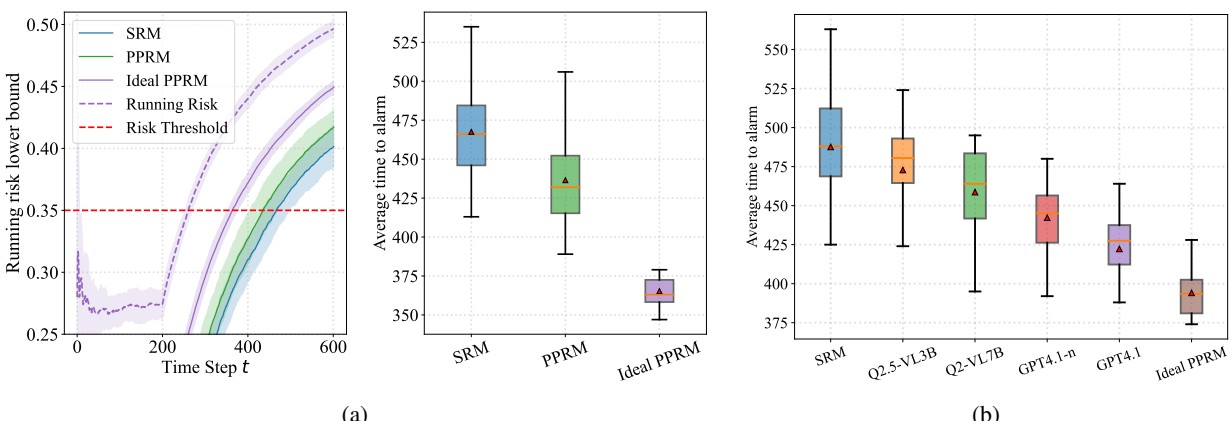

*Figure 8.* Performance for the LLM QA task under prompt shifts: (a) running risk lower bounds and average time to alarm; (b) average time to alarm using different predictors.

## C. Supplementary Experiments

### C.1. Sequential Risk Monitoring on Image Data

In Figure 7(a), we evaluate image classification risk under increasing distribution shift severity, using the squared loss. We observe a consistent performance gain achieved by PPRM. In Figure 7(b), we further consider non-monotonic distribution shift scenarios, where PPRM continues to outperform all benchmark methods.

### C.2. Monitoring an LLM with Limited Human-Labeled Data

In Figures 8(a) and (b), we report additional results for monitoring another lightweight model, namely the 4-bit quantized version of MedGemma-4B-IT (Sellergren et al., 2025), which is specialized for the medical domain. For the predictor $f_p(\cdot)$, we employ the non-specialized model, Qwen2.5-VL-7B in Figure 8(a). The evaluation begins with medical questions and gradually incorporates queries from another domain. The results are consistent with those presented in the main text, demonstrating the generalization capability of PPRM.

### C.3. Detecting the Need for Channel Covariance Re-Estimation

In wireless communication, reliable communication depends on accurate knowledge of the channel statistics, particularly the channel covariance matrix that determines both the quality of *minimum mean-square error* (MMSE) estimation and the performance of downstream decoders. However, in practical wireless systems, the propagation environment evolves over time due to user mobility and temporal variations in interference. These drifts cause a mismatch between the true channel

distribution and the one assumed by the receiver, leading to degraded channel estimates and increased decoding errors. As a result, a central question is when the receiver should trigger covariance re-estimation or retraining of its model parameters. Re-estimating too frequently wastes pilot and computation resources, whereas re-estimating too late leads to significant performance loss. This motivates a principled mechanism that can continuously monitor the channel quality and reliably detect when a statistically significant covariance shift occurs. In this subsection, we examine a communication-inspired setting where the monitored model operates over a channel whose characteristics may drift over time.

### C.3.1. PROBLEM FORMULATION

We consider a standard uplink communication scenario with a *base station* (BS) equipped with $M$ antennas and a single-antenna *user equipment* (UE). We assume block flat-fading channels, where each coherence block contains $\tau_c$ complex-valued samples. We denote by $t$ the index associated with a single coherence block. Each block contains $n_f$ frames for data transmission, with each frame consisting of a codeword encoding a different message. For each frame $i$ in a coherence block $t$, the $M \times 1$ received signal at the BS is given by $v_t[i,j] = h_t s_t[i,j] + e_t[i,j], \quad j = 1, \dots, n_w$, where $n_w$ is the codeword length, $s_t[i,j] \in \mathbb{C}$ is the information-bearing signal transmitted over channel use $j$, which is drawn from an existing constellation $\mathcal{M}$, $h_t$ is the $M \times 1$ channel vector, and $e_t[i,j] \sim \mathcal{CN}(0, \sigma^2 I_M)$ represents *additive white Gaussian noise* (AWGN) with zero mean and covariance $\sigma^2 I_M$.

The channel vectors are assumed to be conditionally Gaussian distributed given a set of parameters $\psi_t$, i.e., $h_t \,|\, \psi_t \sim \mathcal{CN}(0, C(\psi_t))$. The parameter $\psi_t$ accounts for geometric and electromagnetic properties of the propagation environment that change slowly over the coherence interval index $t$. Furthermore, the covariance matrix can be expressed as

$$C_t = C(\psi_t) = \int_{-\pi}^{\pi} g(\theta_t; \psi_t)\, a(\theta_t) a^{\mathrm{H}}(\theta_t)\, d\theta_t, \tag{56}$$

where $g(\theta_t; \psi_t) \geq 0$ is the power density function dictated by the parameters $\psi_t$ and $a(\theta_t)$ denotes the array response vector at the BS for an angle of arrival $\theta_t$.

In order to support the estimation of the channel covariance matrix $C_0$ and the channel noise $\sigma^2$, at the first coherence block, indexed as $t = 0$, the UE sends $n_c \leq \tau_c$ pilots $x_c = [x_c[1], \dots, x_c[n_c]]^{\top} \in \mathbb{C}^{n_c \times 1}$. In order to estimate the parameters $C_0$ and $\sigma^2$, we adopt the positive semidefinite least squares estimate approach introduced in (Dietrich et al., 2006), and the estimated parameters are represented by $\widehat{C}_0$ and $\hat{\sigma}^2$.

At each coherence block $t$, excluding $t = 0$, $n_h$ pilots $x_{t,h} = [x_{t,h}[1], x_{t,h}[2], \dots, x_{t,h}[n_h]]^{\top} \in \mathbb{C}^{n_h \times 1}$ are transmitted to enable channel estimation. We consider MMSE channel estimation, and thus the estimated channel vector $\hat{h}_t$ is given by

$$\hat{h}_t = \widehat{C}_0 Z_{t,h}^{\mathrm{H}} \left( Z_{t,h} \widehat{C}_0 Z_{t,h}^{\mathrm{H}} + \hat{\sigma}^2 I_{M n_h} \right)^{-1} y_{t,h}, \tag{57}$$

where $Z_{t,h} = x_{t,h} \otimes I_M$ and $y_{t,h} = Z_{t,h} h_t + e_{t,h}$. At each block $t$, the BS carries out equalization using the estimated channel vector $\hat{h}_t$ in (57). The MMSE equalized signal $\tilde{s}_t[i,j]$ is obtained as $\tilde{s}_t[i,j] = \left( \hat{h}_t^{\mathrm{H}} \hat{h}_t + \hat{\sigma}^2 \right)^{-1} \hat{h}_t^{\mathrm{H}} v_t[i,j]$. The equalized symbols for each frame are processed by an NN-based decoder, $f(\cdot)$, to obtain an estimate of the transmitted symbol, $\hat{s}_t[i,j] = \arg\max_{m \in \mathcal{M}} f(\tilde{s}_t[i,j])_m$, where $\mathcal{M}$ denotes the modulation set. For risk monitoring, we additionally introduce $n_r$ pilots, $x_{t,r} = [x_{t,r}[1], x_{t,r}[2], \dots, x_{t,r}[n_r]]^{\top} \in \mathbb{C}^{n_r \times 1}$. The equalized pilot signals are denoted as $\tilde{x}_{t,r} = [\tilde{x}_{t,r}[1], \tilde{x}_{t,r}[2], \dots, \tilde{x}_{t,r}[n_r]]^{\top} \in \mathbb{C}^{n_r \times 1}$. These signals are decoded by the decoder, and the decoded pilots are denoted as $\hat{x}_{t,r} = [\hat{x}_{t,r}[1], \hat{x}_{t,r}[2], \dots, \hat{x}_{t,r}[n_r]]^{\top} \in \mathbb{C}^{n_r \times 1}$.

At each coherence block $t \geq 1$, we collect two sets of observations:

- A labeled set from additional pilot symbols: $\mathcal{D}_t = \left\{ (\tilde{x}_{t,r}[k], x_{t,r}[k]) \,\big|\, k = 1, \dots, n_r \right\}$, where $x_{t,r}[k]$ is the ground-truth transmitted pilot symbol.

- An unlabeled set from data symbols: $\tilde{\mathcal{D}}_t = \left\{ \tilde{s}_t[i,j] \,\big|\, i = 1, \dots, n_f, \ j = 1, \dots, n_w \right\}$, for which the true symbol $s_t[i,j]$ is unknown.

Given a decoder $f(\cdot)$ producing the symbol estimate $\hat{s}_t[i,j] = \arg\max_{m \in \mathcal{M}} f(\tilde{s}_t[i,j])_m$, the prediction-powered risk

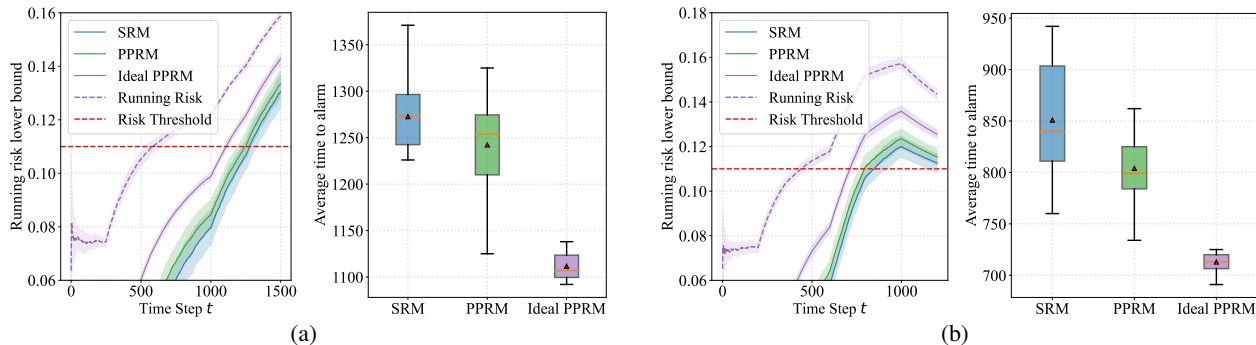

*Figure 9.* Risk estimates as a function of time $t$ and average time to alarm for the channel equalization task: (a) under increasing $\psi_t$ ; (b) under a non-monotonic $\psi_t$.

estimator at block $t$ is given by

$$\hat{R}_t^{\text{PP}} = \underbrace{\frac{\eta_t}{n_w n_f} \sum_{i=1}^{n_f} \sum_{j=1}^{n_w} \ell(f(\tilde{s}_t[i,j]), \hat{s}_t[i,j])}_{\hat{R}_t^{\text{U}}} + \underbrace{\frac{1}{n_r} \sum_{k=1}^{n_r} \Big( \ell(f(\tilde{x}_{t,r}[k]), x_{t,r}[k]) - \eta_t \cdot \ell(f(\tilde{x}_{t,r}[k]), \hat{x}_{t,r}[k]) \Big)}_{\hat{R}_t^{\text{rect}}}. \qquad (58)$$

### C.3.2. RESULTS

We consider a scenario for the simulation in which a BS with 16 antennas receives uplink transmissions from a single-antenna UE. The BS antenna array is assumed to be a uniform linear array, with an inter-element spacing of half the wavelength. The codeword length is set to $n_w = 16$, the number of frames per coherence block is $n_f = 2$ and the signal-to-noise ratio is 6 dB. Additionally, we set $n_c = 500$, $n_h = 3$, and $n_r = 1$. For simplicity, we consider the 3GPP spatial channel model for a uniform linear array with only a single propagation path. In this case, we have only one parameter for the covariance matrix: the angle of the path center $\psi_t \in [-\pi/2, \pi/2]$, which is uniformly distributed. The power density function of the angle of arrival is given by

$$g(\theta_t; \psi_t) = \exp\left( -\frac{d_{2\pi}(\theta_t, \psi_t)}{\sigma_{\text{ASD}}} \right), \qquad (59)$$

where $d_{2\pi}(\theta_t, \psi_t)$ is the wrap-around distance between $\theta_t$ and $\psi_t$, which can be thought of as $|\theta_t - \psi_t|$ for most $(\theta_t, \psi_t)$ pairs. In addition, $\sigma_{\text{ASD}}$ is the angular standard deviation, as it determines how large the deviations from the nominal angle are. We gradually change the parameter $\psi_t$ to simulate an increase in risk. Besides, squared loss is selected as the risk function.

Here we investigate a self-synthetic setting, where no additional models are needed to generate the synthetic labels. The results in Figure 9 demonstrate that PPRM can achieve earlier detection on average compared to SRM. These results highlight PPRM's suitability for on-device model maintenance, where quick adaptation decisions are critical under bandwidth and labeling constraints.

## D. Additional Analysis

### D.1. Effect of Window Length

We analyze how the choice of the window length $L$ affects the performance of PPRM in terms of risk monitoring and detection delay. The results in Figure 10 show that PPRM is relatively robust to the choice of $L$. In addition, the results in the right panel show a clear trade-off. Smaller windows are better for abrupt changes, whereas larger windows exhibit better performance for gradual changes. Across most window sizes, PPRM achieves a shorter average time to alarm than the SRM baselines. These results suggest that a moderate window length offers a good balance between detection sensitivity and stability. According to these results, we select $L = 60$ for its good performance.

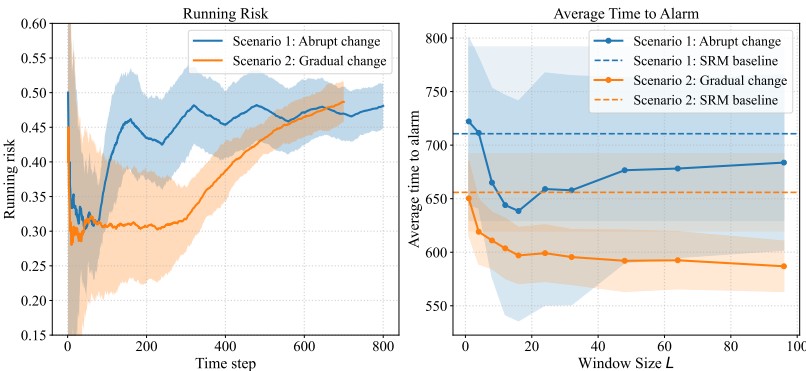

*Figure 10.* Comparison of running risk and average time to alarm under abrupt and gradual change scenarios for an image classification task.

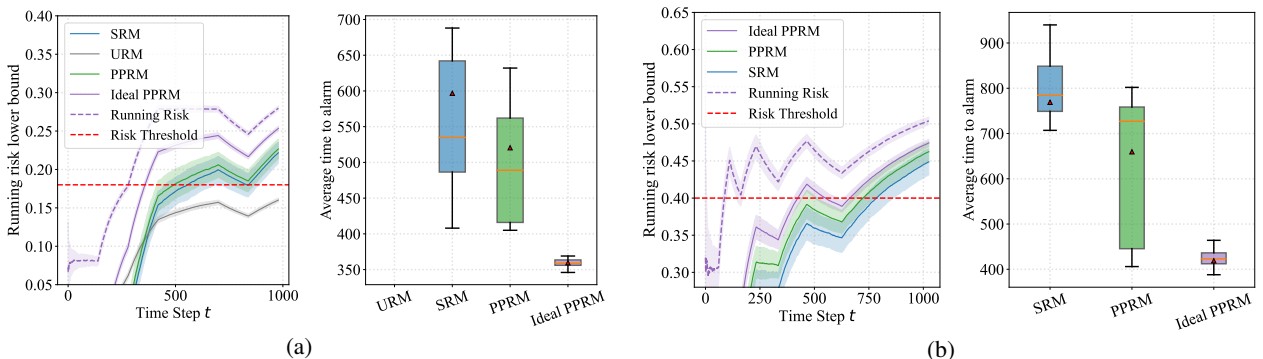

*Figure 11.* Risk estimates as a function of time $t$ under abrupt shifts: (a) for an image classification task; (b) for an LLM QA task.

### D.2. Monitoring the Risk under Abrupt Shifts

In Figure 11, we evaluate the performance of PPRM under abrupt distribution shifts, where the risk suddenly increases at a certain time step. We observe that PPRM can quickly detect the shift and raise an alarm.

### D.3. Effect of Predictor Accuracy

We further analyze how the accuracy of the predictor affects the performance of PPRM in terms of risk monitoring and detection delay. The results in Figure 12 show that as the accuracy of the predictor increases, PPRM achieves a tighter confidence sequence and faster detection compared to SRM. When the predictor is inaccurate, the adaptive version of PPRM performs comparably to SRM, while the non-adaptive PPRM may even underperform SRM due to the use of a poorly calibrated predictor. This demonstrates the benefit of the adaptive mechanism, which prevents PPRM from relying too heavily on an uninformative predictor. Moreover, in the right column of Figure 12, we also report the curve of the obtained $\eta_t$, which is selected by optimizing the variance reduction at each time step. We observe that the optimized $\eta_t$ tends to be larger when the predictor is more accurate.

In Figure 13, we report additional results for the image classification task, using predictors of different accuracies. The results show that PPRM can effectively leverage predictors of varying quality, where a more accurate predictor leads to a tighter confidence sequence and faster detection. Moreover, we also report the curve of cumulative squared prediction error $V_t^{\text{PP}}$ in the right panel of Figure 13. From the result, we can see that the resulting $V_t^{\text{PP}}$ of an accurate predictor is smaller than that of a less accurate predictor, which is consistent with the variance reduction analysis in Appendix A.3.

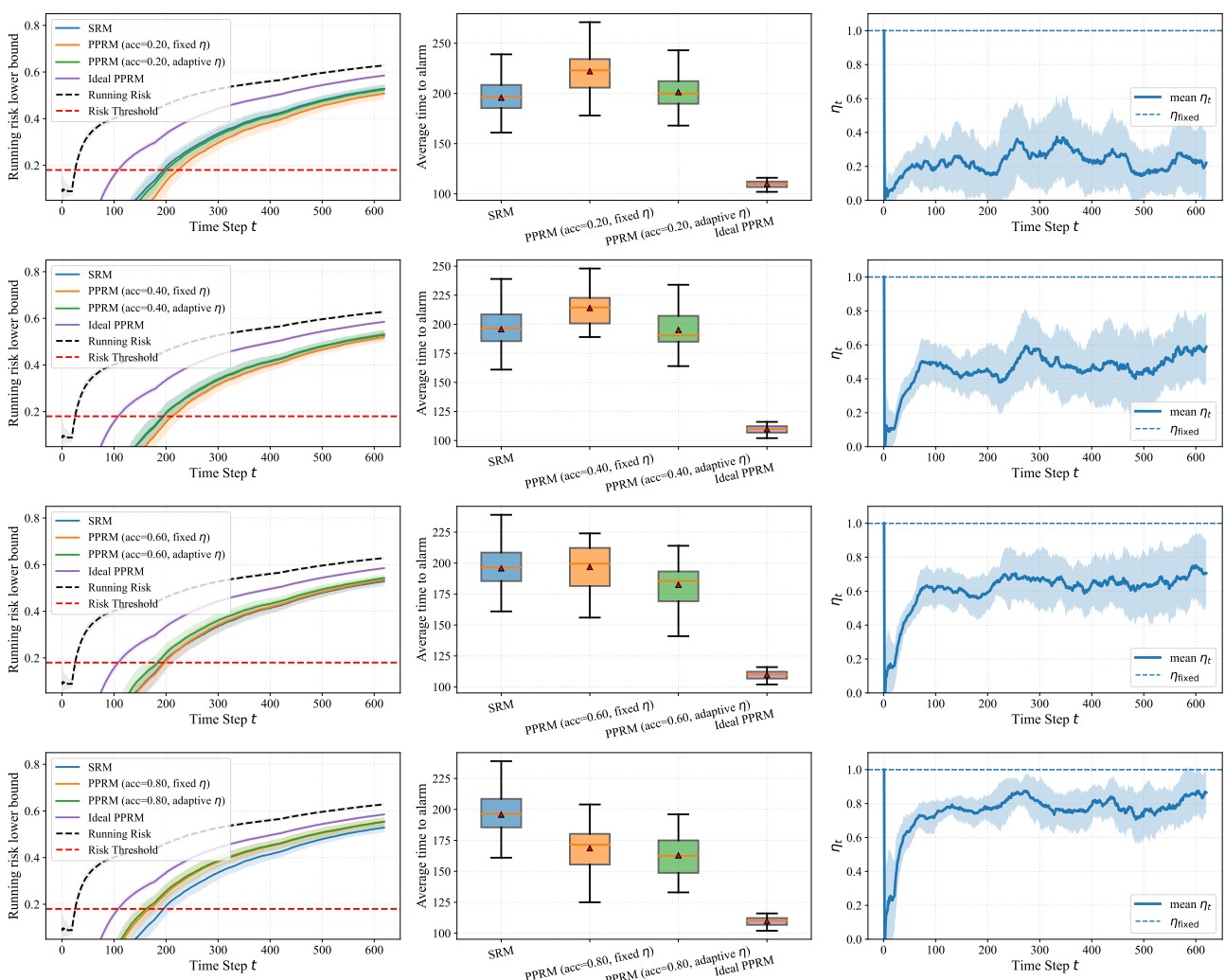

*Figure 12.* Results under increasing shift severity for an image classification task, using predictors of different accuracies.

## D.4. Intermittent Label Streams

In this section, we extend our analysis to the setting where labels are only observed at a subset of time steps, and evaluate the performance of PPRM under this intermittent label stream scenario. Specifically, we consider a setting where labels are observed at every $k$-th time step, i.e., at time steps $t \in \{k, 2k, 3k, \ldots\}$, while at other time steps, only unlabeled data are available. We adapt the PPRM estimator to this setting by only updating the risk estimate at time steps where labels are observed, while using the prediction-powered estimates at all time steps.

We now provide a simple derivation for the intermittent-label setting, in which labels are observed only every $k$ time steps. Let $\mathcal{L}_t = \{k, 2k, \ldots, \lfloor t/k \rfloor k\}$ denote the set of labeled time steps up to time $t$, and let $\mathcal{U}_t = \{1, \ldots, t\} \setminus \mathcal{L}_t$ denote the unlabeled ones. Then, the running risk can be decomposed as

$$\bar{R}_t = \frac{1}{t} \sum_{s=1}^{t} R_s = \frac{1}{t} \sum_{s \in \mathcal{L}_t} R_s + \frac{1}{t} \sum_{s \in \mathcal{U}_t} R_s. \tag{60}$$

For unlabeled time steps $s \in \mathcal{U}_t$, we use the unsupervised lower-bound idea from Appendix B. Specifically, summing over

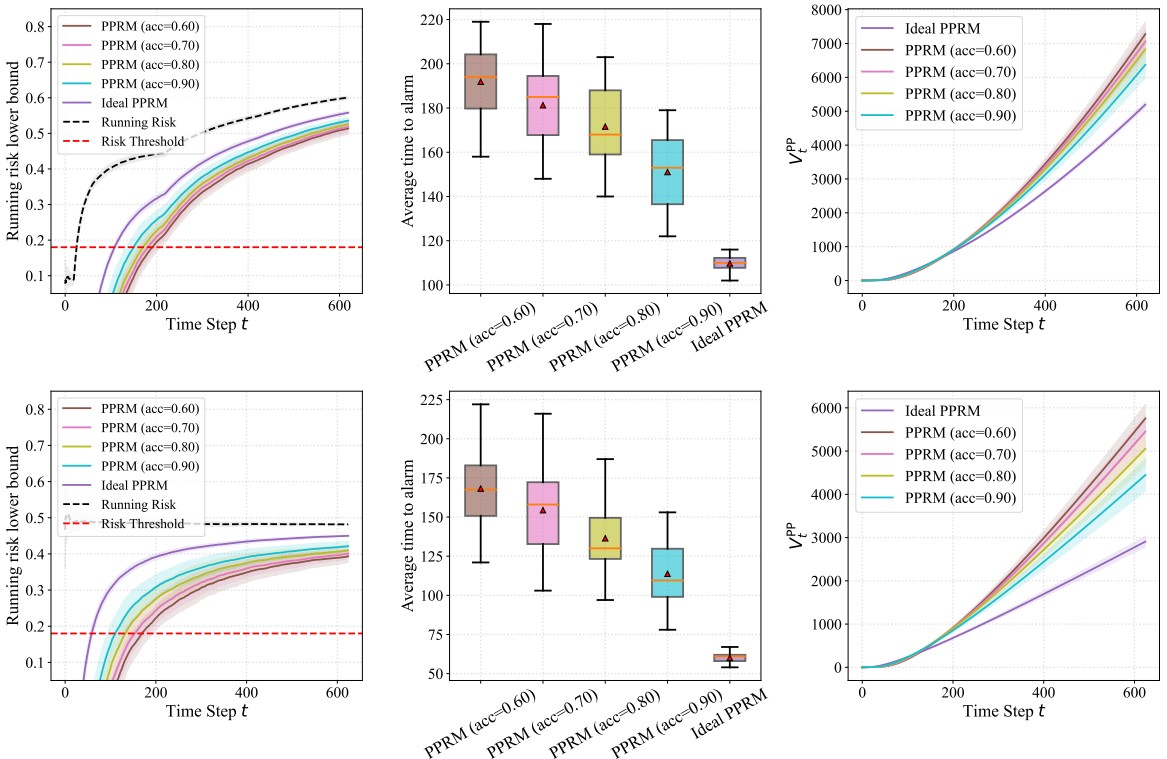

*Figure 13.* Results under increasing shift severity for an image classification task, using different predictors.

all unlabeled time steps gives

$$\frac{1}{t} \sum_{s \in \mathcal{U}_t} R_s \geq \frac{\tau}{t} \left( \sum_{\substack{s=1 \\ s \bmod k \neq 0}}^{t} \mathbb{P}(r_s > \beta_s) - \big(t - \lfloor t/k \rfloor\big)\mathbb{P}_{P_0}(r_0 > \beta_0,\, u_0 \leq \tau) \right), \tag{61}$$

where $\beta_s$ and $\tau$ are the proxy thresholds for the unlabeled time steps, which can be selected by maximizing the F1 score based on the source model's proxy, as in (Schirmer et al., 2025). Combining the labeled and unlabeled terms, we get the following simplified lower bound for the running risk

$$\bar{R}_t \geq \frac{1}{t} \sum_{j=1}^{\lfloor t/k \rfloor} R_{jk} + \frac{\tau}{t} \left( \sum_{\substack{s=1 \\ s \bmod k \neq 0}}^{t} \mathbb{P}(r_s > \beta_s) - \big(t - \lfloor t/k \rfloor\big)\mathbb{P}_{P_0}(r_0 > \beta_0,\, u_0 \leq \tau) \right). \tag{62}$$

Then, we can construct a lower confidence sequence for the running risk by applying the PPRM estimator to the labeled time steps and using the approximation in (Schirmer et al., 2025) to the unlabeled time steps. This expression also makes explicit that the intermittent-label setting interpolates between PPRM and unsupervised risk monitoring. When $k = 1$, all time steps are labeled and (62) reduces to the fully labeled/PPRM monitoring setting after replacing the risks by their PPRM estimates. In contrast, as $k \to \infty$, labeled observations disappear and the bound reduces to a purely unsupervised lower bound based on the proxy scores.

We evaluate the performance of PPRM under this intermittent label stream setting, and compare it with the standard SRM method that only updates the risk estimate at time steps where labels are observed. The results in Figure 14 show that PPRM can still effectively leverage the additional predictors to achieve faster detection compared to SRM.

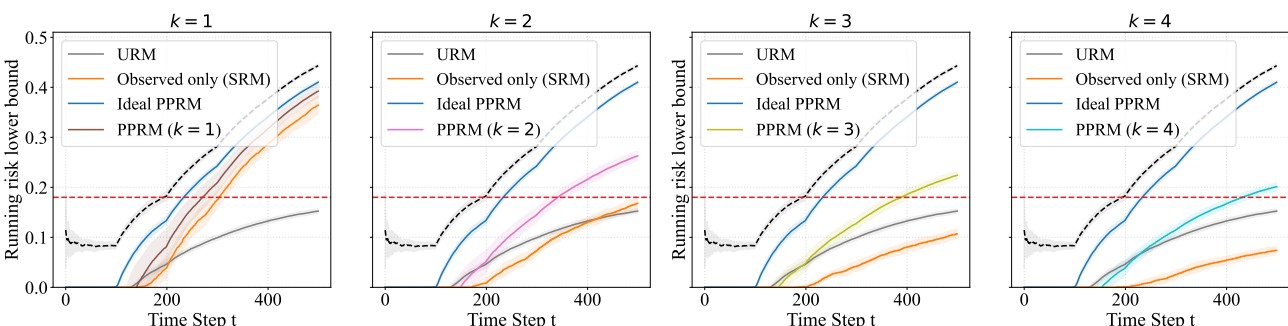

*Figure 14.* Risk estimates as a function of time $t$ under increasing shift severity for an image classification task, in the intermittent label stream setting.

## D.5. Cost Analysis

We further discuss the additional cost introduced by PPRM compared with SRM. One key component of PPRM is the estimation of the adaptive coefficient, which can be computed using simple empirical covariance and variance statistics from observed samples. This estimation is lightweight and incurs negligible overhead. In addition, the auxiliary predictor required by PPRM is flexible and does not necessarily introduce a heavy computational burden. It can be implemented as a lightweight model for efficient online deployment, as a stronger offline model when additional computation is available, or even as the deployed model itself, as demonstrated in our experiments.

