# OpenReview forum: "Prediction-Powered Risk Monitoring of Deployed Models for Detecting Harmful Distribution Shifts"
_ICML.cc/2026/Conference — ICML 2026 regular_

### Official Review · Reviewer_SQyf · 2026-02-26

**Soundness:** 4
**Presentation:** 4
**Significance:** 3
**Originality:** 2
**Overall Recommendation:** 4
**Confidence:** 3

**Summary:**

This paper studies monitoring problem; more specifically, given test datasets time by time with distribution shifts, it raises an alarm if the prediction loss becomes larger than certain threshold. For such a problem, algorithms called SRM (for fully supervised data) and URM (for fully unsupervised data) have been developed. This paper proposes a method called PPRM for semi-supervised data, where the estimation of the loss using semi-supervised data is based on the methodology called prediction-powered inference (PPI). This method can control the probability of false alarm, and under the control it is empirically shown that PPRM can alarm faster than SRM and URM.

**Compliance With Llm Reviewing Policy:**

Affirmed.

**Key Questions For Authors:**

- Overall: (Restatement of "Weaknesses") It looks that the method is built by the combination of SRM and PPI, but what are the originality and/or the difficulty of combining SRM and PPI? The reviewer felt that the combination is very straightforward.
- Lemma 2.1: In the definition $u(V\_t) = \\mathrm{sup}\\{ \\dots \\}$, for what variable is the supremum taken? Perhaps $q$?
  - Related to the above, can the integral easily computed? If so, please provide how to compute it.
- Lemma 3.3: How $u\_t$ and $\\tilde{u}\_t$ are defined? Since both $x\_t$ and $\\tilde{x}\_t$ are sets of samples (not representing single sample), the meaning of $f(x\_t)$ and $f(\\tilde{x}\_t)$ are not obvious. Perhaps $f(x\_t) = [ f(x\_{t,i}) ]_{i=1}^{n\_t}$?
  - Also, the computation of (26) is weird since the size of $x\_t$ and $\\tilde{x}\_t$ are different and therefore the covariance between $u\_t$ and $\\tilde{u}\_t$ cannot be defined. As far as reading (Angelopoulos et al., 2023b), should the second argument of "Cov" be $\\ell(f(x\_t), f\_p(x\_t))$ (not by unlabeled data but by labeled data)?
- Section 5: Can the probability of the false alarms (that the proposed method controls) in the experiments presented?
- Section 5: In the experiments the proposed method was compared only with SRM and URM, but can it be compared with other methods (like methods in Section 4)? If difficult, why?
- Section 5.2: Although the procedure of data distribution changes is presented for image data (Section 5.1), it is not presented for LLM experiment. What was the procedure?

---- **The followings are minor questions**

- Section 2.1: Just curiosity: In this paper it controls the probability of false alarms and under the control it aims to alarm earlier. Can it be modified for controlling the probability of un-alarmed case (even the risk exceeds the threshold) and under the control reducing the number of alarms?
- Section 4: Although URM is used in the experiment (Section 5), it is not discussed in Section 4. Why? Perhaps is it because the methodology is quite different?

**Limitations:**

Yes

**Strengths And Weaknesses:**

Strengths:

- [significance] Extension of existing methods for supervised and unsupervised settings to semi-supervised settings is not only theoretically but also practically useful.
- [soundness][significance] By carefully building the method, the probability of the false alarm can be controlled based on the prediction loss for true test distribution (not just empirical test distribution).
- [significance][presentation] Figure 4 is interesting; this method requires a model to label unlabeled data, and the performance by the choice of the model is examined.

Weaknesses:

- [originality] It looks that the method is built by the combination of SRM and PPI, but the originality and/or the difficulty of combining SRM and PPI are unclear.
- [significance] Experiment is conducted only with the method the proposed method based on (SRM and URM), which may be insufficient.

---

> ### Author Rebuttal · Authors · 2026-03-30
>
> Thanks for your comments!
>
> **Weaknesses**:
> - W1: On the originality
>   - First, we formulate a new sequential semi-supervised risk monitoring problem: unlike existing works, our setting allows each time step to contain both a small labeled batch and a larger unlabeled batch.
>   - Second, we construct prediction-powered estimators of the running risk and show that, when the reliance parameter $\eta_t$ is predictable, the resulting monitoring rule preserves finite-sample false-alarm control. This first adapts prediction-powered inference from static estimation to sequential risk monitoring.
>   - Third, our framework can use the deployed model itself to generate synthetic labels, which broadens applicability in resource-constrained settings.
>   - Fourth, beyond standard image classification, the paper studies LLM-based monitoring and a telecommunications application, showing that the formulation is relevant across distinct domains.
> - W2: On the empirical evaluation
>   - Our current experiments cover a range of diverse and practically relevant settings, including (i) telecommunications tasks, (ii) real-world datasets for image classification, and (iii) LLM-based monitoring scenarios.
>   - Meanwhile, we have included a number of new experiments in this [**link**](https://anonymous.4open.science/r/icmlr-53EC).
>
> **Questions**:
> - Q1: On the originality
>   - Please refer to our replies to W1.
> - Q2: Definition of the supremum
>   - We thank the reviewer for pointing this out. The supremum is taken over the scalar variable $S$, not over $x$. For a fixed $V_t$, $u(V_t)$ is defined as the largest $S$ such that
> 	$$\int_0^1 q(\lambda)\exp \big(\lambda S-\psi_E(\lambda)V_t\big)\,d\lambda < \frac{1}{\delta_T}
> 	$$for commonly used mixture distributions $q(\lambda)$, such as Gamma distributions, the above integral can be computed analytically, which makes the computation of bound efficient in practice.
> - Q3: On the Lemma 3.3
>   - In our formulation, $u_t = \ell(f(x_t), y_t)$ and $\tilde{u}_t = \ell(f(\tilde{x}_t), f_p(\tilde{x}_t))$ are defined at the population level as random variables corresponding to a single draw from the labeled and unlabeled distributions, respectively. Hence, $u_t$ and $\tilde{u}_t$ should be interpreted as generic sample-level losses, not set-valued quantities.
>   - The covariance in Eq. (26) is defined with respect to the underlying joint distribution of labeled and unlabeled samples, and is computed at the sample level. Also, here we indeed need to use different notations. An expanded expression of Eq. (26) is $$\eta_t^*=\frac{\mathrm{Cov}\big(\ell(f(x\_t), y\_t),\ell(f({x}\_t), f\_p({x}\_t))\big)}{\left(1 + \frac{n\_t}{N\_t}\right)\mathrm{Var}\big(\ell(f(\tilde{x}\_t), f_p(\tilde{x}\_t)) \big)}.$$
>   - We confirm that the second term in covariance should indeed take the form $f_p$, i.e., $\ell(f(x_t), f_p(x_t))$. We will fix this problem.
> - Q4: On the false alarm probability
>   - We report the false alarm probability in this [**link**](https://anonymous.4open.science/r/icmlr-53EC/falarm.png). It is shown that false alarm probability can be effectively controlled.
> - Q5: On the benchmarks
>    - In Section 5, we focused our empirical comparison on SRM and URM because they represent the two most directly relevant baselines for our setting. Both are also explicitly designed for the same sequential risk monitoring objective as our proposed PPRM. We have also unified the URM and PPRM formulations and included more simulations in intermediate scenarios, please see our replies to *Reviewer Y3Uz*.
>    - The additional methods discussed in Section 4 mostly fall into related but not directly comparable categories. For example, several approaches target different hypotheses, i.e., instantaneous risk rather than running risk, assume distinct experimental setups, e.g., full unsupervised settings or test-time adaptation. As a result, adapting them to our problem formulation would require nontrivial modifications.
>    - We agree that broader empirical comparisons would be valuable. We will clarify this rationale in the revised paper.
>    - We include more experiment settings, which can be found in this [**link**](https://anonymous.4open.science/r/icmlr-53EC).
> - Q6: On the LLM settings
>   - In the LLM experiments, we vary the datasets at each timestep to induce distribution shifts, and evaluate performance under these time-varying distributions. We will revise Section 5.2 to explicitly describe this.
> - Q7: On the new formulations
>   - In principle, this would require a different formulation of the sequential testing objective. Such guarantees require additional assumptions, such as additional information on the risk.
> - Q8: On the introduction of URM
>   - Yes, URM is of a different methodology. We have actually introduced it in Appendix B.

---

> > ### Author Rebuttal · Reviewer_SQyf · 2026-04-04
> >
> > Thank you for rigorous responses. I was convinced except for the following point.
> >
> > Additional question: Regarding Question 1, my main concern is not the points in the response, but the point "The reviewer felt that the combination is very straightforward". For this point, how the authors think?

---

> > > ### Author Response · Authors · 2026-04-04
> > >
> > > Thank you for your suggestions and feedback. PPRM is constructed by integrating SRM and PPI, while ensuring that the resulting procedure implements a valid sequential monitor. This integration is not straightforward. For example, a direct approach, e.g., replacing the empirical risk in SRM with a PPI estimate, would not yield the anytime-valid guarantee required in our setting. This is due to the following reasons.
> > > - First, PPI was originally developed for static inference, whereas sequential risk monitoring requires an anytime-valid sequential lower bound. Establishing such a guarantee requires a rigorous new analysis.
> > > - Second, the prediction-powered estimator is not in the standard bounded form that SRM can handle directly. We therefore need to introduce an affine normalization together with a modified variance process.
> > > - Third, a simple integration of PPI with SRM does not automatically lead to the performance gains reported in our paper. This is illustrated in the table below, where *average time to alarm* is reported.
> > >
> > >   | Predictor Accuracy (%) | SRM | PPRM (Fixed) | PPRM (Adaptive) |
> > >   |:-:|:-:|:-:|:-:|
> > >   |20|196|222|200|
> > >   |40|196|214|195|
> > >   |60|196|196|182|
> > >   |80|196|168|162|
> > >
> > >   PPRM (Fixed) represents an initial design and only helps in sufficiently accurate regimes.
> > > - This motivates our adaptive variance-reduction design. The resulting form is similar to that in Angelopoulos et al. (2023b), which we cite directly because the problem formulation is similar. However, choosing the parameter $\eta_t$ using data observed at time $t$ would generally destroy predictability, introducing bias and invalidating the statistical guarantee. This is why we design the past-dependent estimator in Section 3.3. In our setting, validity is preserved only when the reliance parameter $\eta_t$ is predictable, which is exactly why Lemma 3.1 and Theorem 3.2 are needed.
> > >
> > > Beyond this, we have conducted a range of new experiments and explored self-imputed strategies for label generation. Moreover, we will provide more in-depth analysis (please refer to the reply to *Q1 of Reviewer fh9g*), and will also make an effort to unify PPRM with related frameworks such as URM (please refer to the reply to *W2 of Reviewer Y3Uz*).
> > >
> > > To avoid overstating the contribution, we will revise the paper to explicitly say that the method is conceptually transparent, but that the main contribution is proving that this combination is sequentially valid and adaptive in the semi-supervised setting.

---

### Official Review · Reviewer_L36E · 2026-03-10

**Soundness:** 2
**Presentation:** 2
**Significance:** 2
**Originality:** 1
**Overall Recommendation:** 4
**Confidence:** 3

**Summary:**

The core issue of this article is to explore the problem where machine learning models deployed in dynamic environments suffer from changes in data distribution due to environmental variations, thereby affecting the model's performance. The author focuses on monitoring harmful distribution changes because they can lead to risks in model operation exceeding the normal level. To address the challenge of scarce labeled feedback in actual deployment, this article proposes a semi-supervised monitoring framework based on predictive-driven reasoning, namely Prediction-Powered Risk Monitoring (PPRM).

This method utilizes unlabeled data to generate synthetic labels and estimates model risk through an auxiliary predictor. The bias correction term derived from labeled data ensures that the result estimator remains unbiased. This framework can construct any valid lower bound for the running test risk, and triggers an alarm when the lower bound exceeds the threshold, while reducing the possibility of false alarms. Experimental results on tasks such as image classification, LLM, and telecommunications have shown that PPRM can detect harmful distribution changes earlier than existing supervised or unsupervised methods.

**Compliance With Llm Reviewing Policy:**

Affirmed.

**Final Justification:**

The paper is technically sound and presents an interesting application of PPI to monitoring. Given the improved clarity on theoretical conditions and the additional evidence on robustness, I have rised my score accordingly(4: Weak Accept). It is suggested that the authors provide a clearer explanation in the revised manuscript.

**Key Questions For Authors:**

1.This method relies on synthetic labels. So, how would its performance be when the predictor has significant deviations?

2.Theoretical guarantee requires that the hyperparameter ηₜ should form a predictable sequence that solely depends on historical observations. Then, how can this constraint be ensured when using an adaptive tuning strategy?

3.Considering the computational costs and latency impacts of invoking LLM, how should we optimize and address these issues in the three scenarios presented in this article?

4.In formula 1 of the text, there is an assumption that the samples are independent and identically distributed. However, in actual data streams, there may exist unstable and dynamic patterns caused by time variations. So, how robust is the proposed method when this assumption is violated?

**Limitations:**

Yes

**Strengths And Weaknesses:**

Strength

1.This article discusses the monitoring of harmful distribution changes, and it is particularly applicable to scenarios where labels are continuously generated and the annotation process is both costly.

2.The PPRM theory proposed in this paper is very solid. By combining predictive inference with continuous risk monitoring, it has conducted thorough experiments on risk levels and average alarm events.

3.This paper has verified the universality of PPRM across various fields, and all of them have demonstrated stable monitoring performance.

Weakness

1.It is relatively limited in terms of innovation. The supervised risk monitoring and prediction-powered inference have been integrated into a semi-supervised monitoring process.

2.The empirical assessment is relatively limited in scale. Moreover, the prompt words used in the LLM monitoring experiments were affected.

3.The formula 20 in Lemma 3.1 is used to ensure the predictability dependent on , which refers to the historical data. Since this paper is discussing a dynamic environment, the theoretical proof at the current moment does not provide this.

4.This framework relies on the quality of the synthetic labels, but the paper does not analyze the robustness when the predictor is inaccurate.

---

> ### Author Rebuttal · Authors · 2026-03-30
>
> Thanks for your comments!
>
> **Weaknesses**:
> - W1: On the innovation
>   - First, we formulate a new sequential semi-supervised risk monitoring problem: unlike prior supervised monitoring methods that require labeled test data at every step, and unsupervised approaches that rely on stronger assumptions on proxy quality, our setting allows each time step to contain both a small labeled batch and a larger unlabeled batch.
>   - Second, we construct prediction-powered estimators of the running risk and show that, when the reliance parameter $\eta_t$ is predictable, the resulting monitoring rule preserves finite-sample false-alarm control. This first adapts prediction-powered inference from static estimation to *sequential* risk monitoring.
>   - Third, our framework can use the deployed model itself to generate synthetic labels, which broadens applicability in resource-constrained settings.
>   - Fourth, beyond classic image classification, the paper studies LLM-based monitoring and a telecommunications application, showing that the formulation is relevant across distinct domains. We have unified the URM and PPRM, and included more simulations in intermediate scenarios, please see our replies to *Reviewer Y3Uz*.
> - W2: On the empirical evaluation
>   - We covered a range of diverse settings, including (i) telecommunications tasks, (ii) real-world image classification datasets, and (iii) LLM-based monitoring scenarios. These are intended to demonstrate the generality of our framework across different domains. In response, we have conducted and released additional experiments and settings, please find these figures [**here**](https://anonymous.4open.science/r/icmlr-53EC). We will integrate these new results in the revised version.
>   - We emphasize that our method treats LLM outputs as black-box predictions. Therefore, the theoretical guarantees of PPRM hold regardless of how these predictions are generated, including the specific prompt design. To improve reproducibility, we will include the exact prompts used in our experiments and add further analysis on prompt variations.
> - W3: On the guarantees
>   - The key requirement in Lemma 3.1 is predictability. In particular, Eq. (20) relies on the assumption that the hyperparameters form a predictable sequence, meaning they depend solely on historical observations. Under this condition, the unbiasedness result holds irrespective of how the data distribution evolves over time. Thus, even in the presence of distributional shifts, Eq. (20) remains valid, since our construction ensures that all quantities are computed using only past information.
> - W4: On the robustness
>   - We intentionally avoid relying on highly powerful models in our main experiments to reflect practical settings. Nevertheless, Figure 4 shows that stronger predictors can further improve performance, suggesting that PPRM can naturally benefit from better auxiliary models when available. We manually vary the predictor accuracy for more specific investigations here: please see [**case1**](https://anonymous.4open.science/r/icmlr-53EC/cp_1.png) and [**case2**](https://anonymous.4open.science/r/icmlr-53EC/cp_2.png). Consistent observations with Figure 4 can be obtained.
>   - We provide a theoretical analysis in the reply to Q1 of *Reviewer fh9g*. With higher covariance, i.e., prediction accuracy, the performance gain would be larger.
>
> **Questions**:
> - Q1: On the robustness
>   - Please refer to our replies to W4.
> - Q2: On the implementation of predictable sequence
>   - The predictability requirement means that $\eta_t$ can only depend on past observations. This is satisfied in our implementation, since $\eta_t$ is computed using historical data up to time $t-1$, e.g., via the sliding-window estimator in Eq. (27), and is therefore independent of the current data at time $t$.
> - Q3: Computational cost and latency
>   - LLMs are only used in the LLM-based scenario; the other two tasks do not involve LLMs. In the LLM task, the computational cost and latency can be optimized by choosing a proper model size and applying inference acceleration techniques. In practice, the resulting cost is typically lower than that of obtaining human-annotated labels. For non-LLM tasks, the predictor $f_p$ can be a lightweight model or even the deployed model itself, as demonstrated in our experiments.
> - Q4: On the i.i.d. assumption within a batch
>   - In Eq. (1), we assume data in a batch are i.i.d.  When samples within a batch are dependent but share the same marginal distribution, the target risk remains well-defined. However, dependence typically increases the variance of the empirical risk estimator by reducing the effective sample size. Thus, concentration-based thresholds may become overly optimistic, which can degrade detection performance. However, we need to emphasize that i.i.d. assumption is widely accepted, and remains effective in most cases. Moreover, we do not assume independent data across different batches in different time steps.

---

> > ### Author Rebuttal · Reviewer_L36E · 2026-04-01
> >
> > Thank you to the authors for the detailed rebuttal. The responses clarify several points and improve the presentation of the paper.
> >
> > W1: I appreciate the clarification that the paper formulates a sequential semi-supervised risk monitoring problem, which adds conceptual novelty. However, I am still not fully convinced that the overall methodological innovation is strong enough to change my original recommendation.
> > W2: The authors’ response suggests that the empirical study is reasonably broad and covers multiple domains. This concern is largely alleviated.
> > W3&Q2: The clarification on how the predictability requirement for  is enforced in implementation is clear. This concern is largely resolved.
> > W4&Q1: My concern was mainly about robustness under weak or biased auxiliary predictors. The response mostly shows that stronger predictors help, so this concern is only partially addressed.
> > Q3: The response is reasonable, but practical deployment involves not only labeling cost, but also latency, system complexity, and extra model calls. This point remains only weakly addressed.
> > Q4: The clarification on the IID assumption is clear and helpful. I appreciate the authors’ transparency on this point.
> >
> > Overall, the rebuttal improves the clarity of the paper and resolves some of my theoretical concerns. While my concerns about the level of novelty, robustness under weak auxiliary predictors, and practical deployment considerations are only partially addressed, the rebuttal provides enough clarification to improve my recommendation. Therefore, I will raise my overall recommendation to 4: Weak Accept.

---

> > > ### Author Response · Authors · 2026-04-01
> > >
> > > We sincerely appreciate the reviewer’s comments and suggestions, and thanks for the positive feedback.
> > >
> > > **On methodological novelty**
> > >
> > > We thank the reviewer for acknowledging the conceptual contribution. We respect the reviewer’s perspective. We will revise the paper to more clearly and concisely highlight the contributions:
> > > - the role of PPRM within the sequential testing framework,
> > > - its distinctions from prior approaches.
> > >
> > > **On robustness**
> > >
> > > We agree that robustness under weak auxiliary predictors is very important.
> > >
> > > In fact, when the auxiliary predictor has very low accuracy, the synthetic labels can introduce negative effects, and naively incorporating them may be harmful. To further investigate this scenario, we conduct additional experiments on the *average time to alarm* with relatively weak predictors, as shown in the table below.
> > >
> > > | Predictor Accuracy (%) | SRM | PPRM (Fixed) | PPRM (Adaptive) |
> > > |:-:|:-:|:-:|:-:|
> > > |20|196|222|200|
> > > |40|196|214|195|
> > > |60|196|196|182|
> > > |80|196|168|162|
> > >
> > > Notably, the monitored model itself maintains an average accuracy of around 50% throughout the entire process.
> > > As shown, when using a fixed $\eta_t = 1$, PPRM may lose its advantage over SRM when the auxiliary predictor accuracy falls below 50%. However, this scenario also highlights a key strength of our adaptive approach. With the proposed **adaptive design of $\eta_t$**, the method is able to detect the limited utility of the auxiliary predictor. In such cases, $\eta_t$ is quickly reduced, effectively ignoring synthetic labels. A more detailed illustration can be found at this [**link**](https://anonymous.4open.science/api/repo/icmlr-53EC/file/acc_v.png?v=64a1d6db).
> > >
> > > The proposed method therefore degrades gracefully to behavior comparable to SRM under weak predictors, while still benefiting from auxiliary predictors when they are informative.
> > >
> > > **On practical deployment**
> > >
> > > Indeed, in real-world deployment scenarios, additional factors such as latency, system complexity, and extra model calls need to be carefully considered beyond labeling cost alone. These practical factors naturally introduce additional design considerations, including the selection of auxiliary predictors, as well as trade-offs between cost and monitoring efficiency. We would like to emphasize that our general framework is flexible with respect to these choices. We will explicitly discuss these practical considerations in the revision.
> > >
> > > Thanks again for the suggestions and inspiration!

---

### Official Review · Reviewer_fh9g · 2026-03-13

**Soundness:** 3
**Presentation:** 3
**Significance:** 3
**Originality:** 2
**Overall Recommendation:** 5
**Confidence:** 4

**Summary:**

This paper studies how to monitor the performance of deployed machine learning models when only a small number of labels are available online. The authors propose Prediction-Powered Risk Monitoring (PPRM), which uses an auxiliary predictor to generate synthetic labels for unlabeled data and then corrects the resulting bias using a small labeled set. The resulting estimator is used within a sequential monitoring framework to detect harmful increases in model risk while maintaining false alarm guarantees. Experiments on image classification, LLM-based QA monitoring, and a telecommunications task suggest that the proposed method can often detect performance degradation earlier than purely supervised monitoring approaches.

**Compliance With Llm Reviewing Policy:**

Affirmed.

**Final Justification:**

This paper studies the problem of monitoring model performance over time and proposes a prediction-powered risk monitoring (PPRM) approach that leverages auxiliary predictors to reduce the variance of risk estimates and enable earlier detection of distribution shifts. The paper is generally well written, and the method is conceptually simple and practically relevant, especially in modern settings where predictions from auxiliary models are readily available. The empirical results across multiple settings, including LLM monitoring scenarios, provide evidence that the proposed approach can lead to earlier detection compared to baseline monitoring strategies.

In the rebuttal and revised manuscript, the authors clarified the intuition behind the method and provided additional discussion illustrating how variance reduction from prediction-powered estimators can lead to faster detection. These clarifications addressed several of my questions about the mechanism of improvement and improved the overall presentation of the work.

Some limitations remain. In particular, the theoretical characterization of detection delay improvements remains somewhat limited, and the empirical improvements over the baselines are sometimes modest. Nevertheless, the rebuttal helped clarify the method and strengthened my confidence in the contribution.

Taking these updates into account, I have increased my score by one point.

**Key Questions For Authors:**

Question 1: The main empirical advantage of the proposed method is earlier detection compared to SRM. While the paper proves unbiasedness and false alarm control, it does not theoretically characterize the detection delay improvement. Could the authors discuss whether such results might be possible under simplified settings (e.g., under Gaussian or other parametric assumptions)? For example, can one theoretically analyze when the prediction-powered estimator leads to earlier alarms compared to SRM?
Question 2: The empirical results compare several auxiliary predictors and show that stronger predictors tend to lead to earlier detection. Could the authors further discuss how predictor quality affects monitoring performance more systematically, for example under controlled settings (e.g., varying predictor accuracy) or when predictor quality changes over time?
Question3: The experiments mainly focus on gradually increasing or non-monotonic shifts. Could the authors comment on how the method behaves under more abrupt distribution changes?

**Limitations:**

yes

**Strengths And Weaknesses:**

Strengths
The paper addresses a well-motivated problem of monitoring deployed ML models when labeled data are scarce. The proposed method integrates prediction-powered estimators with a sequential monitoring framework to leverage unlabeled data while preserving statistical guarantees such as unbiasedness and false alarm control. The experiments include both vision and LLM-based monitoring tasks, and the idea of using LLM predictors in this setting is interesting.
Weakness
the theoretical analysis mainly establishes statistical validity but does not characterize when or why the method should detect shifts earlier than existing approaches. The empirical results suggest that the method benefits from stronger auxiliary predictors, but this dependency is not analyzed in depth. In addition,  the empirical improvements over baselines are relatively modest.

---

> ### Author Rebuttal · Authors · 2026-03-30
>
> Thanks for your comments!
>
> **Weaknesses**:
> - W1: On the rationale and improvements
>   - PPRM employs a prediction-powered estimator that remains unbiased while leveraging unlabeled data through bias correction. Since the stopping rule depends on when the lower confidence bound crosses the threshold, variance reduction directly tightens the confidence sequence and enables earlier detection. Please refer to our replies to your Q1 and Q2 for details.
>   - We agree that a full characterization of detection delay is currently missing. In the revision, we will explicitly highlight this mechanism. Please also refer to our responses to your Q1 and Q2.
>   - Regarding improvements, we deliberately use moderate-quality predictors to better reflect realistic scenarios. As shown in Figure 4, using stronger predictors can improve performance.
>
> **Questions**:
> - Q1: On deeper analysis
>   - In the revision, we will add an example. Under a simplified Bernoulli setting, one can theoretically characterize why PPRM leads to earlier detection. Consider the detection starting from a harmful scenario, where distribution shifts start at $t=1$. Assume the true error rate satisfies
>   $$u\_{t,i} \sim \mathrm{Bernoulli}(p), \quad p > \theta, \quad \Delta = p - \theta > 0,
>   $$where $\theta$ denotes the threshold. Let the synthetic predictions satisfy$$
>   \tilde u\_{t,j} \sim \mathrm{Bernoulli}(q), \quad
>   \tilde u'\_{t,i} \sim \mathrm{Bernoulli}(q),
>   $$
>   and define $\gamma = \mathrm{Cov}(u\_{t,i}, \tilde{u}'\_{t,i})$.
>   The SRM estimator has variance
>   $$v^{\mathrm S} = \frac{p(1-p)}{n},$$ while the PPRM estimator with a certain $\eta$ has variance $$v^{\mathrm{PP}}(\eta) = \eta^2\frac{q(1-q)}{N} + \frac{1}{n}\Big(p(1-p)+\eta^2 q(1-q)-2\eta \gamma\Big). $$
>   For large $t$, the estimator concentrates around $p$, i.e.,$\hat{\bar{R}}\_t^{\mathrm{PP}} \approx p.$ Hence, since $V\_t^{\mathrm{PP}}=\sum_{s=1}^t(\hat{R}\_s^{\mathrm{PP}}-\hat{\bar{R}}\_{s-1}^{\mathrm{PP}})^2$, we have approximation using law of large numbers, $V_t^{\mathrm{PP}} \approx t v^{\mathrm{PP}}(\eta)$.
>   Since observations are bounded, we adopt a sub-exponential $\psi_{E,1}$.
>   Using the linear boundary from Lemma 1 (Howard et al., 2021), this yields the approximate detection delay
>   $$ T_{\mathrm{PP}}^{\mathrm{avg}}(\lambda,\eta)\approx \frac{(1+2\eta)^2\log(1/\delta_T)}{\lambda(1+2\eta)\Delta-\psi_{E,1}(\lambda) v^{\mathrm{PP}}(\eta)},$$ For SRM, the delay is
>   $$T_{\mathrm S}^{\mathrm{avg}}(\lambda) \approx\frac{\log(1/\delta_T)} {\lambda\Delta-\psi_{E,1}(\lambda)v^{\mathrm S}}.$$ Therefore, PPRM is better than SRM whenever
>   $$v^{\mathrm{PP}}(\eta)<(1+2 \eta)^2 v^{\mathrm{S}}-\frac{2 \eta(1+2 \eta) \lambda \Delta}{\psi\_{E, 1}(\lambda)}.$$In particular, PPRM alerts earlier when $\gamma$ is larger, i.e., using more accurate predictor.
> - Q2: On accuracy of auxiliary predictor
>   - The current version already contains two pieces of evidence in this direction. First, in the LLM experiment, using progressively stronger predictors consistently reduces the average time to alarm (Figure 4). Second, Table 1 shows that adaptive tuning of $\eta_t$ is most beneficial when the predictor is relatively weak, which shows that the advantage of PPRM depends on how informative the synthetic labels are. We add the following experiments: (i) manually controlled predictor-quality experiment and (ii) variance–accuracy relationship.
>   Results are shown in [**link 1**](https://anonymous.4open.science/r/icmlr-53EC/cp_1.png) and [**link 2**](https://anonymous.4open.science/r/icmlr-53EC/cp_2.png), where higher accuracy brings better performance.
>   - Moreover, we have naturally accounted for time-varying predictor quality, since we employ practical predictors whose accuracy changes over time.
> - Q3: On diverse distribution shifts
>   - We would like to clarify that our method does not rely on any assumption of gradual or smooth change. The false-alarm guarantee established in Theorem 3.2 holds for arbitrary sequences of distributions satisfying the null hypothesis, and thus remains valid even under abrupt or sudden shifts.
>   - The primary impact of abrupt shifts is on detection efficiency rather than validity. Additional experiments on the average time to alarm under more abrupt shifts are provided below.
>
>     |Method|SRM|PPRM|Ideal PPRM|
>     |:-:|:-:|:-:|:-:|
>     |LLM|925|880|771|
>     |Image Classification|559|521|358|
>
>     Detailed results for LLM and image classification tasks can be found in [**link 1**](https://anonymous.4open.science/r/icmlr-53EC/abrupt_1.png) and [**link 2**](https://anonymous.4open.science/r/icmlr-53EC/abrupt_2.png), respectively. Our proposed strategy remains effective in these settings. We further investigated the impact of window size $L$. Smaller $L$ is better when abrupt shifts are involved. Results are shown in this [**link**](https://anonymous.4open.science/r/icmlr-53EC/windowsize.png).

---

> > ### Author Rebuttal · Reviewer_fh9g · 2026-04-02
> >
> > Thank you for the detailed rebuttal and for sharing the updated manuscript.
> >
> > The response clarifies the intuition behind the proposed PPRM approach, particularly how prediction-powered estimators can reduce variance and potentially lead to earlier detection. The additional discussion and the simplified example help illustrate the mechanism through which PPRM may outperform SRM. I also appreciate the additional experimental results and clarifications provided in the revised version.
> >
> > While the theoretical characterization of detection delay improvements remains somewhat limited and the empirical improvements over baselines appear relatively modest, the rebuttal helps clarify the method and strengthens the overall presentation of the work.
> >
> > Taking these updates into account, I have increased my score by one point.

---

> > > ### Author Response · Authors · 2026-04-03
> > >
> > > Thank you very much for your careful reading of our paper and rebuttal. We are particularly grateful for your insightful comments regarding the theoretical characterization of detection delay improvements. In the revised version, we will expand this discussion and provide a more complete and rigorous derivation to better substantiate the claimed improvements.
> > >
> > > Thanks again for your valuable suggestions!

---

### Official Review · Reviewer_Y3Uz · 2026-03-24

**Soundness:** 3
**Presentation:** 2
**Significance:** 2
**Originality:** 2
**Overall Recommendation:** 4
**Confidence:** 4

**Summary:**

The paper studies risk monitoring in semi-supervised setting by leveraging the framework of prediction powered inference. The idea is use an auxiliary model to impute the labels that are unavailable at test time for estimating the risk. However, such imputation procedure can be provably corrected by assuming access to some small amount of labelled data using the prediction powered inference. The framework is tested on image classification, LLM monitoring on increasingly difficult or complex prompts over time. The comparison is made against supervised version of monitoring, and the recently proposed unsupervised version with experiments demonstrating the ability of the proposal to track the true risk as well as on the stopping times to raise the alarm.

**Compliance With Llm Reviewing Policy:**

Affirmed.

**Final Justification:**

The paper is sound for the problem it is trying to solve. It is an interesting application of PPI for monitoring. However, the evaluation could have been bit more tight as right now it is among methods that are are truly faithful to the approach, for example comparison to supervised, and unsupervised, while the work is semi-supervised. The authors have tried to acknowledge and address some of such concerns. Hence, I'm not against this paper getting accepted.

**Key Questions For Authors:**

1. I do not understand the citation of Zawalski et al. 2025 for the argument that data distribution may shift over time. Clearly this problem has been studied for almost a decade now, and there is a lot to cite here. So this one-off recent 2025 citation seems odd (to me). Curious to hear the reasoning.
2. I think PFA as an acronym for probability of false alarm is misleading. PFA stands for please find attached. While this is not a major concern, but probability of false alarm has already common name like type-I error control or false alarm control. I'd be cautious of renaming or reinventing acronyms for already known terms.
3. In equation 8, what is the expectation over?
4. Citation style is not consistent in the paper.
5. In related work, the paper compares to the approaches monitoring or providing false-alarm guarantees on the instantaneous risk. Could it be clarified beyond the change in the hypotheses being that instead of tracking running or time-averaged risk to using expected loss at each time step, how else is the hypotheses across frameworks are different.

**Limitations:**

yes

**Strengths And Weaknesses:**

Strengths:
The core idea is sound, and interesting as well. Application of PPI to reliably estimate the risk of the deployed model is also a nice application. The paper is generally well-written. The paper is also sound.

Weaknesses:
1. Some of my concerns are with the empirical setup or the evaluation of the framework. The two main baselines are the SRM and the URM, and I'm not sure if the comparison is justified. URM is designed for the fully unlabelled setting, so comparing PPRM against it, in my opinion, says nothing about whether PPI procedure is helpful. It could simply reflect the value of having the labels. While I do agree that URM guarantee holds under certain assumptions, so PPRM is still a nice methodological approach. However, that also means that these PPRM and URM are two different settings, without fair basis for comparison.
2. For the SRM, I reckon the gains of PPRM comes from variance reduction due to the PPI procedure. This is a reasonable and interesting argument. However, the experimental setup undermines how signficant this benefit actually is. The paper has $n_t =1$ for labelled points per time step. While this is limited, by the time the alarm has been raised (around $>500$ time steps), the monitor has accumulated around $500$ labelled samples. This is not label-limited setting. I might be mis-understanding something here though. It might be interesting to explore the setting where the labels are genuinely scarce, like arriving after $k$ steps.
3. For the comparison against SRM, looking at the stopping times, the difference between SRM and PPRM is not that significant, about 5%-6% reduction. While this is crucial in consequential applications, however it further demands some cost-benefit analysis of SRM and PRM (like the complexity of estimating $\eta_t$, and the requirement of $f_p$).
4. Furthermore, some concerns remain with $\eta_t$. I'm not sure I understood the goal of this adaptive procedure. The sliding window estimates $Cov(u_t, \tilde{u}_t)$ and the variance from past data under different distributions, yet the benefit of adaptation relies on these estimates being correct under the current distribution $P_t$. Under fast or abrupt shifts, the windowing mechanism is not probably the most reliable. I would be curious to understand this more.

---

> ### Author Rebuttal · Authors · 2026-03-30
>
> Thanks for your comments!
>
> **Weaknesses**:
>
> - W1: On comparison with URM
>   - Our main objective is to explore strategies in a semi-supervised setting. We agree that directly comparing PPRM with URM may be unfair. Inspired by your W2, we formulate a more general setting where labels arrive every $k$ steps. This perspective allows us to place both URM and PPRM within a unified framework, where URM and PPRM can be viewed as two extreme cases.
> - W2: On label-limited cases
>   - Since the shift occurs at $t=200$ and the threshold is crossed at $t=280$, only post-shift labels are informative for detection, rather than all labels. Thus, the actual number of useful labels is smaller.
>   - The intermittent case you mentioned is highly interesting. Under an assumption similar to Eq. (39), we can derive the following simplified bound:
>   $$\bar{R}\_t \geq \frac{1}{t} \sum_{j=1}^{\lfloor t / k\rfloor} \hat{R}\_{j k}^{\mathrm{PP}}+\frac{\tau}{t}\left(\sum\_{\substack{t^{\prime}=1, t^{\prime} \bmod k \neq 0}}^t \mathbb{P}\left(u\_{t^{\prime}}>\lambda_{t^{\prime}}\right)-(t-\lfloor t / k\rfloor) \mathrm{PFP}\_0\right).$$
>   Then, we can construct a valid $L_t$ for detection. This formulation makes clear that PPRM and URM arise as special cases: when $k=1$, every time step is labeled and we get PPRM; when $k\to\infty$, we recover URM. Here we provide results on this by varying $k$:
>
>     |$k$|1 (PPRM)|2|3|4|$\infty$ (URM)|
>     |:-:|:-:|:-:|:-:|:-:|:-:|
>     |Average time to alarm|271|344|387|432|-|
>
>   For more results, please refer to this [**link**](https://anonymous.4open.science/r/icmlr-53EC/intm.png).
> - W3: On improvements and efficiency
>   - In our experiment, we deliberately avoid using powerful models in order to better reflect real-world scenarios. As shown in Figure 4, using stronger predictors can improve performance.
>   - We need to clarify that the mentioned 5–6% improvement may be based on a metric that underestimates the gain. A more suitable measurement is the *average detection delay (ADD)* (Gary Lorden) defined as the additional detection steps after the true risk crosses the threshold. Using this, the relative improvement would increase to approximately 20% and even higher.
>   - The adaptive coefficient $\eta_t$ is estimated via simple empirical covariance and variance statistics, which is negligible. Moreover, $f_p$ can be implemented as a lightweight model, an offline stronger model, or even the deployed model itself, as demonstrated in our paper.
> - W4: On adaptive $\eta_t$
>   - The purpose of the adaptive rule is not to estimate current distribution perfectly, but to choose a predictable coefficient that approximately minimizes the one-step prediction error. The adaptive method is designed to approximate Eq. (26). Since it is not observable, we estimate it from past data using a sliding window.
>   - We agree that we implicitly assume a certain level of temporal stability. However, two points are worth clarifying: the statistical validity of PPRM does not depend on the accuracy of this estimate for $\eta_t$; and adaptation mainly affects detection speed, not reliability. An intuitive example can be found in our replies to Q1 and Q2 of *Reviewer fh9g*.
>   - Average time to alarm results on more abrupt shifts are given below
>
>     |Method|SRM|PPRM|Ideal PPRM|
>     |:-:|:-:|:-:|:-:|
>     |LLM|925|880|771|
>     |Image Classification|559|521|358|
>
>     Results for LLM and image classification tasks are shown in [**link 1**](https://anonymous.4open.science/r/icmlr-53EC/abrupt_1.png) and [**link 2**](https://anonymous.4open.science/r/icmlr-53EC/abrupt_2.png), respectively. Our proposed strategy remains effective in these settings. We also investigated the impact of window size $L$. Results are shown in this [**link**](https://anonymous.4open.science/r/icmlr-53EC/windowsize.png). Smaller $L$ is sometimes better when abrupt shifts are involved.
>
> **Questions**:
> - Q1: On the citation
>   - Our intention in citing Zawalski et al. (2025) was not only to position it as a reference on distribution shift, but also to highlight a new example where machine learning systems are deployed in dynamic environments.
> - Q2: On terminology (PFA)
>   - We will use false-alarm control.
> - Q3: On Eq. (8)
>   - The expectation in Eq. (8) is taken over the distribution of the first alarm time, i.e., averaging over the randomness of the entire data streams.
> - Q4: On citation style
>   - We will revise this.
> - Q5: On differences with instantaneous methods
>   - Our method tests for persistent degradation relative to a baseline, i.e., $\bar R_t \le R_0 + \epsilon_{\mathrm{tol}}$. In contrast, (Timans et al., 2025; Shekhar & Ramdas 2023; Csillag et al., 2025) test pointwise safety constraints $R_t \le \tau$ at each time step.
>   - We target persistent performance degradation, whereas instantaneous risk monitoring methods are designed to detect instantaneous violations.
>   - Moreover, most instantaneous risk monitoring methods only studied supervised data streams.

---

> > ### Author Rebuttal · Reviewer_Y3Uz · 2026-04-04
> >
> > Thanks for the rebuttal. My concerns are resolved. I'll update my score a bit. The paper still need some work on the clarity and presentation.

---

> > > ### Author Response · Authors · 2026-04-04
> > >
> > > Thank you very much for your suggestions! We appreciate your feedback and will further improve the clarity and presentation in the revision.

---

### Decision · Program_Chairs · 2026-04-30

**Decision:**

Accept (regular)

**Comment:**

This paper proposes a PPI-variant of the risk monitoring framework originally proposed by (Podkopaev & Ramdas, 2021).  Unlabeled data is combined with labeled data to reduce variance, with the core contribution being how to introduce PPI without violating the anytime-valid guarantee.  The reviewers had favorable opinions of the work (3 x weak accept, 1 x accept), with the major concerns being about presentation / clarity and the novelty of combining PPI with risk monitoring and whether adding PPI results in a clear improvement in the detection delay (theoretically and/or empirically).  I believe the authors have addressed these concerns, especially in the latter case by adding the Bernoulli example.